# Blockchain-based isotopic big data-driven tracing of global PM sources and interventions

Yuming Huang[1,2], Xiangyu Li[1], Yuehan Wu[1], Chaoyang Xue [3], Jiashuo Li [4], Yongfeng Lin [1], Wei Nie [5], Xian Liu [1], Qian Liu [1], Greg Michalski[6], Jingwei Zhang [7] ✉, Zheng Zong [8] ✉, Dawei Lu [1,9] ✉ & Guibin Jiang[1]

Tracing sources and assessing intervention effectiveness are crucial for controlling atmospheric particulate matter (PM) pollution. Isotopic techniques enable precise top-down tracing, but the absence of long-term, global-scale multi-compound isotopic data limits comprehensive analysis. Here, we establish a blockchain-based isotopic database, compiling 34,815 isotopic fingerprints of global PM and its emissions from 1,890 pollution events across 66 countries. This allows retrospective analysis and predictions, revealing that PM sources are distinct, dynamically changing over time, and often asynchronous with interventions. Additionally, we estimate source contributions to $PM_{2.5}$ and its compounds, highlighting the increasing impact of biomass burning. Furthermore, projections indicate that by 2100, PM levels may decline to $5.38 \pm 0.16\ \mu g/m^3$ in the Americas and $13.9 \pm 1.82\ \mu g/m^3$ in Asia under climate mitigation scenarios but will still exceed WHO guidelines without further controls on natural emissions. Guiding future interventions with isotopic big data is essential for addressing air pollution challenges.

Atmospheric particulate matter (PM) pollution significantly impacts Earth's radiation budget[1], global climate[2], and public health[3,4]. Exposure to ambient PM, even at low concentrations, harms human health[5]. Consequently, the World Health Organization (WHO) updated the recommended annual $PM_{2.5}$ (PM ≤ 2.5 μm in diameter) exposure limit to 5 μg/m³ in 2021[6]. However, recent global estimates indicate that 99.82% of the land area and 99.99% of the population are still exposed to $PM_{2.5}$ levels exceeding this limit[6]. This underscores PM pollution as a pressing global challenge, necessitating effective interventions.

Precisely tracing PM emissions and evaluating the effectiveness of past pollution interventions are essential for designing future PM strategies. However, it is difficult because PM composition is complex, including both local emissions and regional transport[7]. The pollution levels exhibit a complex nonlinear relationship with emissions. This introduces uncertainties in previous bottom-up emission inventories and molecular-/element-based top-down tracing methods[8]. Recently, isotopic techniques have provided high-precision fingerprints for top-down tracing of PM emissions[9]. Isotope ratios tend to show distinct

[1]Key Laboratory of Environmental Chemistry and Toxicology, Research Center for Eco-Environmental Sciences, Chinese Academy of Sciences, Beijing 100085, China. [2]Sino-Danish College, Sino-Danish Center for Education and Research, University of Chinese Academy of Sciences, Beijing 100049, China. [3]Max Planck Institute for Chemistry, Mainz, Germany. [4]Institute of Blue and Green Development, Shandong University, Weihai 264209, China. [5]Joint International Research Laboratory of Atmospheric and Earth System Research, School of Atmospheric Sciences, Nanjing University, Nanjing, China. [6]Department of Earth, Atmospheric, and Planetary Sciences, Purdue University, 550 Stadium Mall Drive, West Lafayette, IN, USA. [7]Yunnan Key Laboratory of Meteorological Disasters and Climate Resources in the Greater Mekong Subregion, Yunnan University, Kunming 650500, China. [8]Environment Research Institute, Shandong University, Qingdao, Shandong 266237, China. [9]Hubei Key Laboratory of Environmental and Health Effects of Persistent Toxic Substances, School of Environment and Health, Jianghan University, Wuhan 430056, China. ✉e-mail: jwzhang@ynu.edu.cn; zzong@sdu.edu.cn; dwlu@rcees.ac.cn

signatures (i.e., fingerprints) among various sources due to different formation processes[10]. Importantly, these isotopic fingerprints are preserved through complex atmospheric processes[11]. To date, 14 elemental isotopes, carbon ($f_{M-14C}$ for radioactive; $\delta^{13}C$ for stable), nitrogen ($\delta^{15}N$), oxygen ($\delta^{18}O$), silicon ($\delta^{30}Si$), sulfur ($\delta^{34}S$), iron ($\delta^{56}Fe$), nickel ($\delta^{60}Ni$), copper ($\delta^{65}Cu$), zinc ($\delta^{66}Zn$), strontium ($\delta^{87}Sr$), neodymium ($\delta^{144}Nd$), hafnium ($\delta^{177}Hf$), lead ($^{207}Pb/^{206}Pb$), and mercury ($\delta^{202}Hg$), have been utilized for source tracing of PM species[12–15]. Nevertheless, these applications mainly concentrate on short-term PM pollution events on a local scale. Only two global long-term nitrogen isotope studies have focused on nitrates and ammonium[12,16], which is inadequate for assessing overall PM. Therefore, a global long-term analysis of PM and their emissions' isotopic big data is particularly important for tracing PM sources and interventions.

To achieve global isotopic big data analysis, we developed the Isotopic Database for Global Atmospheric Research (IDGAR). A blockchain architecture was integrated into IDGAR to ensure the accuracy and traceability of isotopic big data. Based on IDGAR, we created a comprehensive global isotopic map of 1890 pollution events across 66 countries from 1957–2023, revealing the regional-specific source-end isotopic values for PM tracing. Moreover, we identified main emissions and assessed their specific effective intervention durations for different PM species by analyzing the trends and change rates of isotopic big data. Relying on these comprehensions, we estimated the source composition of $PM_{2.5}$ and its critical components, and characterize the spatiotemporal variations of these source composition. We reveal the rising impact of biomass burning and liquid fossil fuels combustion. Furthermore, we project future $PM_{2.5}$ trends in scenarios aimed at meeting climate goals of 1.5 °C and 2 °C by 2100. The blockchain-based isotopic big data directly connects PM, sources, and interventions, providing important guidance for future atmospheric research.

## Results

### Construction of IDGAR by using blockchain technology

We developed the IDGAR database (http://idgar.org/) by compiling 34,815 non-redundant global isotopic observations, including 7,053 source and 27,762 PM isotopic data published between 1957 and 2024 (Supplementary Fig. 1)[17]. These observations include 14 elements across 1,890 PM pollution events and isotopes such as $f_{M-14C}$, $\delta^{13}C$, $\delta^{15}N$, $\delta^{18}O$, $\delta^{30}Si$, $\delta^{34}S$, $\delta^{56}Fe$, $\delta^{60}Ni$, $\delta^{65}Cu$, $\delta^{66}Zn$, $\delta^{87}Sr$, $\delta^{144}Nd$, $\delta^{177}Hf$, $^{207}Pb/^{206}Pb$, and $\delta^{202}Hg$ (Supplementary Fig. 1). For light elements (e.g., C, N, S), the isotopic characteristics of different compounds, including organic carbon (OC), elemental carbon (EC), nitrate ($NO_3^-$), ammonium ($NH_4^+$), and sulfate ($SO_4^{2-}$), have been thoroughly compiled. Moreover, we integrated a blockchain architecture into IDGAR's data management system, ensuring that the isotopic-driven tracing process is immutable and traceable.

Specifically, the curated isotopic data and their details (e.g., sampling date, site location, reference), along with user operations (including authentication information, downloads, and uploads), are transformed into unique 64-character hexadecimal strings using the Secure Hash Algorithm (SHA256, a cryptographic function) and stored as a block in IDGAR (Supplementary Fig. 1)[18–20]. The SHA256 function is one-way and unbreakable, making the transformed isotopic data tamper-proof. Moreover, each future user action or data update is digitally recorded with timestamps, which are input into the SHA256 function along with the previous block's string to generate a new block string. Through this iterative hashing pattern, each block is closely linked to all preceding blocks in chronological order, thereby systematically growing the blockchain. Even a minor alteration results in distinct changes in subsequent blocks, enabling precise identification of affected blocks and files by tracing back to the initial block that experienced a hash change. By ensuring data tamper-proofing and increasing trust, IDGAR facilitates the accurate and sustainable

application of publicly shared atmospheric isotopic big data for researchers worldwide.

### Global isotopic map of PM sources

To obtain complete and definitive source-end isotopic values for accurately explaining PM origins, we developed the global isotopic map of PM sources based on IDGAR. This map includes the source isotopic fingerprints of various PM species, e.g., EC, OC, $NO_3^-$, $NH_4^+$, $SO_4^{2-}$, and metals, across diverse global regions (from 75.10° S to 83.20° N) between 1957–2023 (Figs. 1a and Supplementary Fig. 2). As shown in Fig. 1, different PM species tend to show different source information. For instance, EC has sources such as coal combustion, C3 plants burning emissions, C4 plants burning emissions, and vehicle exhaust, while OC is not entirely the same, as it also includes emissions from liquefied petroleum gas. Therefore, PM source tracing require multi-species isotopic fingerprints. Notably, most isotopic fingerprints showed notable variations among these sectors for different species ($P < 0.05$), indicating their potential for PM source distinction (Fig. 1b-m and Supplementary Fig. 3). Moreover, a closer examination revealed regional heterogeneity in source isotopic fingerprints of certain individual species, e.g., Pb, across Asia, Europe, and the Americas ($P < 0.05$, Supplementary Table 1 & 2). This variability may be attributed to the diverse industrial structures and lifestyles in these regions. However, formal bootstrap testing indicated no statistical dependence of isotopic fingerprints on latitude and longitude within these regions ($P > 0.05$, Supplementary Fig. 4 and Supplementary Note 1). These blockchain-based continent-specific source isotopic values enable to provide accurate reference for tracing PM and their species sources.

### Main sources and interventions for different PM species

IDGAR provides an unprecedented opportunity for long-term analyzes of global PM isotopic fingerprints, enabling the identification of shifts and intervention effectiveness on main sources of PM species. The isotopic fingerprints of PM species represent a linear mixture of different sources. By analyzing their temporal trends and comparing them with the aforementioned isotopic fingerprints, the main sources during a corresponding period can be identified. If the rates of change in PM isotopic fingerprints (i.e., the slope between isotopic fingerprints and year) show a continuous decline but do not reverse direction, it indicates that the main sources' contributions are increasing, but the growth rate is slowing down. This suggests that the intervention targeting the main sources has begun to take effect. Perhaps this scenario is suspected to be due to the increase in contributions from sources with opposite isotopic compositions. However, if this is the case, the isotopic fingerprints of PM will show opposite trend changes and the corresponding change rates will also reverse direction. Therefore, analyzing the temporal trends and change rates of isotopic big data can identify the main sources of PM species, and evaluate the effectiveness and timing of interventions on these main sources.

Considering the temporal trends and magnitudes may be influenced by the selection of the start and end years, we performed formal bootstrap testing procedures and moving subset window analysis. We found that global $f_{M-14C}$, $\delta^{15}N$, $^{207}Pb/^{206}Pb$, $\delta^{18}O$, $\delta^{30}Si$, $\delta^{34}S$, $\delta^{56}Fe$, $\delta^{65}Cu$, $\delta^{66}Zn$, $\delta^{87}Sr$ and $\delta^{202}Hg$ of PM species show varying temporal dynamics, including different trend breaks, tendencies, and change rates (Fig. 2a–h, Supplementary Figs. 5, 6). The robustness of the statistical results was verified using the Theil–Sen estimator (Supplementary Figs. 7, 8). This indicates that different PM components have distinct main sources and interventions during different periods. For instance, the global $f_{M-14C}$ of EC in PM increased significantly before 2014, then declined (Fig. 2a), indicating that their main emissions shift from non-fossil fuel source (i.e., biomass burning, $f_{M-14C} = 1$) to fossil fuel combustion (coal or oil, $f_{M-14C} = 0$)[21,22] in 2014. By analyzing the EC $\delta^{13}C$ trend and source isotopic values, we further identified whether

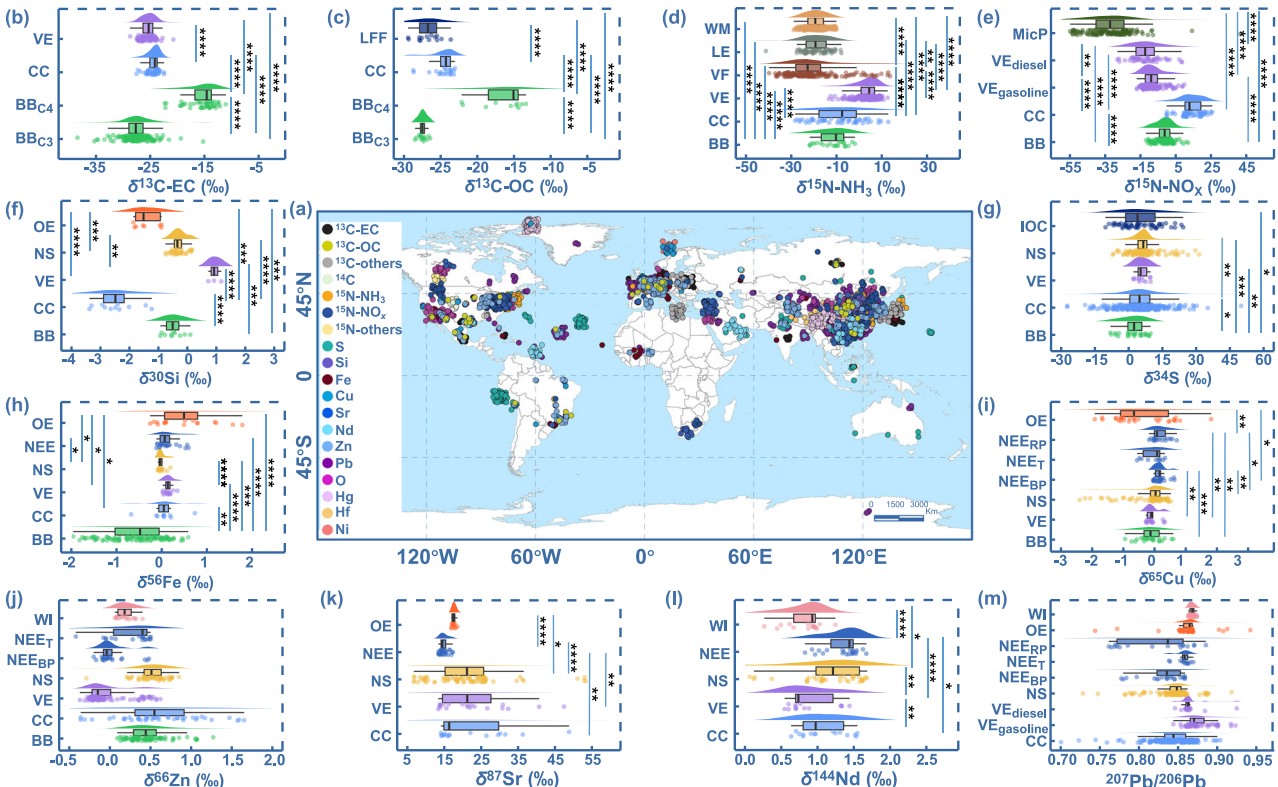

**Fig. 1 | Isotopic map and statistical results of global atmospheric particulate matter (PM) sources. a** Geographical distribution of 7,053 isotopic fingerprints in global PM emissions. The map was created using Natural Earth. Free vector and raster map data are available at naturalearthdata.com. **b–m** Statistical differences in isotopic fingerprints among different emissions. The labels VE, CC, BB, BB$_{C4}$, BB$_{C3}$, LFF, WM, LE, VF, MicP, VE$_{diesel}$, VE$_{gasoline}$, OE, NS, IOC, NEE, NEE$_{RP}$, NEE$_{T}$, NEE$_{BP}$, and WI in (**b–m**) represent vehicle exhausts, coal combustion, biomass burning, C4 plants burning, C3 plants burning, liquid fossil fuels combustion, waste materials, livestock emissions, volatilized fertilizer, microbial processes, diesel vehicle emissions, gasoline vehicle emissions, ore-related emissions, natural soil, industrial oil combustion, non-exhaust emissions, non-exhaust emissions-road paint, non-

exhaust emissions-tire, non-exhaust emissions-brake pad and waste incinerator. The isotopic composition of ammonium (NH$_4^+$) and nitrate (NO$_3^-$) sources is conventionally expressed in terms of their precursor compounds, namely the isotopic composition of NH$_3$ and NOx, respectively[47,48]. The box extends from the 25% to the 75%, the center line represents the median, and the whiskers indicate the 5% and 95% of the data points. Significance levels are indicated as follows: \*\*\*\**P* < 0.0001, \*\*\**P* < 0.001, \*\**P* < 0.01, \**P* < 0.05. *Note:* the isotopic compositions of Zn (**j**) and Pb (**m**) from different sources show significant differences (*P* < 0.05), but there are too many statistical results to be clearly labeled in the figure. Therefore, all the data are provided in the Source Data file. Additionally, details on the number of isotopic data points in each window (**b–m**) are also provided in the Source Data file.

biomass burning before 2014 was dominated by C4 or C3 plants. Specifically, the rise in EC $\delta^{13}$C during 2001–2008 and 2012–2014 suggests C4 plants burning ($\delta^{13}$C-EC = −15.4 ± 2.73‰) was the main source, while the decline from 2009–2011 points to C3 plants burning ($\delta^{13}$C-EC = -27.5 ± 2.69‰). Notably, the rate of increase in $f_{M-14C}$ slowed continuously during 2005–2007, and 2012–2014 (Fig. 2b), suggests effective interventions targeting C4 plants emissions during these periods (Supplementary Fig. 9).

In contrast to EC, global $f_{M-14C}$ of OC in PM dropped sharply before 2015, then rose (Fig. 2c), indicating fossil fuel combustion (oil or coal) dominated pre-2015, while biomass burning ($f_{M-14C}$ = 1) became dominant afterward. The OC $\delta^{13}$C has steadily increased, surpassing the isotopic end-member value for oil (Supplementary Figs. 5a and Fig. 1c), pointing to coal combustion as the main source before 2015, with a shift to C4 plants emissions thereafter. The slower decline in $f_{M-14C}$ of OC from 2009 to 2011 (Fig. 2d) suggests effective interventions on coal combustion during this period (Supplementary Fig. 9).

Moreover, global $\delta^{15}$N of ammonium salts in PM decreased from 2001 to 2013, then rose from 2014 to 2020 (Fig. 2e), indicating a shift in main sources of ammonium salts from isotope-depleted to isotope-enriched ones, e.g., vehicle exhaust. Notably, during 2017–2020, the rate of $\delta^{15}$N-NH$_4^+$ increase slowed continuously (Fig. 2f), suggesting effective interventions targeting vehicle-derived NH$_4^+$-bearing PM (Supplementary Fig. 9). In contrast, $\delta^{15}$N of NO$_3^-$ increased from

−12.0 ± 15.9‰ to 6.1 ± 6.5‰ between 2001 and 2018, suggesting coal combustion as the main source. Afterward, $\delta^{15}$N-NO$_3^-$ declined, pointing to non-coal sources, e.g., gasoline combustion, microbial processes, or biomass burning (Fig. 2g). The rate of $\delta^{15}$N-NO$_3^-$ increase slowed between 2013 and 2018 (Fig. 2h), indicating effective measures against coal-related NO$_3^-$ in PM (Supplementary Fig. 9). Similarly, $\delta^{34}$S in sulfate generally declined, with a brief increase from 2015 to 2016 (Supplementary Fig. 5g). The rate of decline slowed during 2010–2012 and after 2020 (Supplementary Fig. 5h). Based on source $\delta^{34}$S values, we suggest that before 2015, coal or biomass combustion was the main source of sulfate, effectively controlled between 2010 and 2012. After 2020, isotopic values fell below 2‰, indicating that biomass burning ($\delta^{34}$S = 1.84 ± 5.41‰) became the dominant source and has been progressively managed.

In addition to these high-abundance components, a detailed discussion on the main sources of other components and the effectiveness of their interventions is provided in the Supplementary Note 2. Overall, effective interventions were only observed for the main sources of PM components, e.g., EC, OC, NO$_3^-$, NH$_4^+$, SO$_4^{2-}$, Pb, Sr, and Hg, and their effectiveness was limited to specific periods rather than the entire study duration (Supplementary Fig. 9). These results indicate that the main sources and corresponding interventions are not synchronized. The heterogeneity of different PM species underscores the importance of considering multi-species isotopes in tracing overall PM source and interventions.

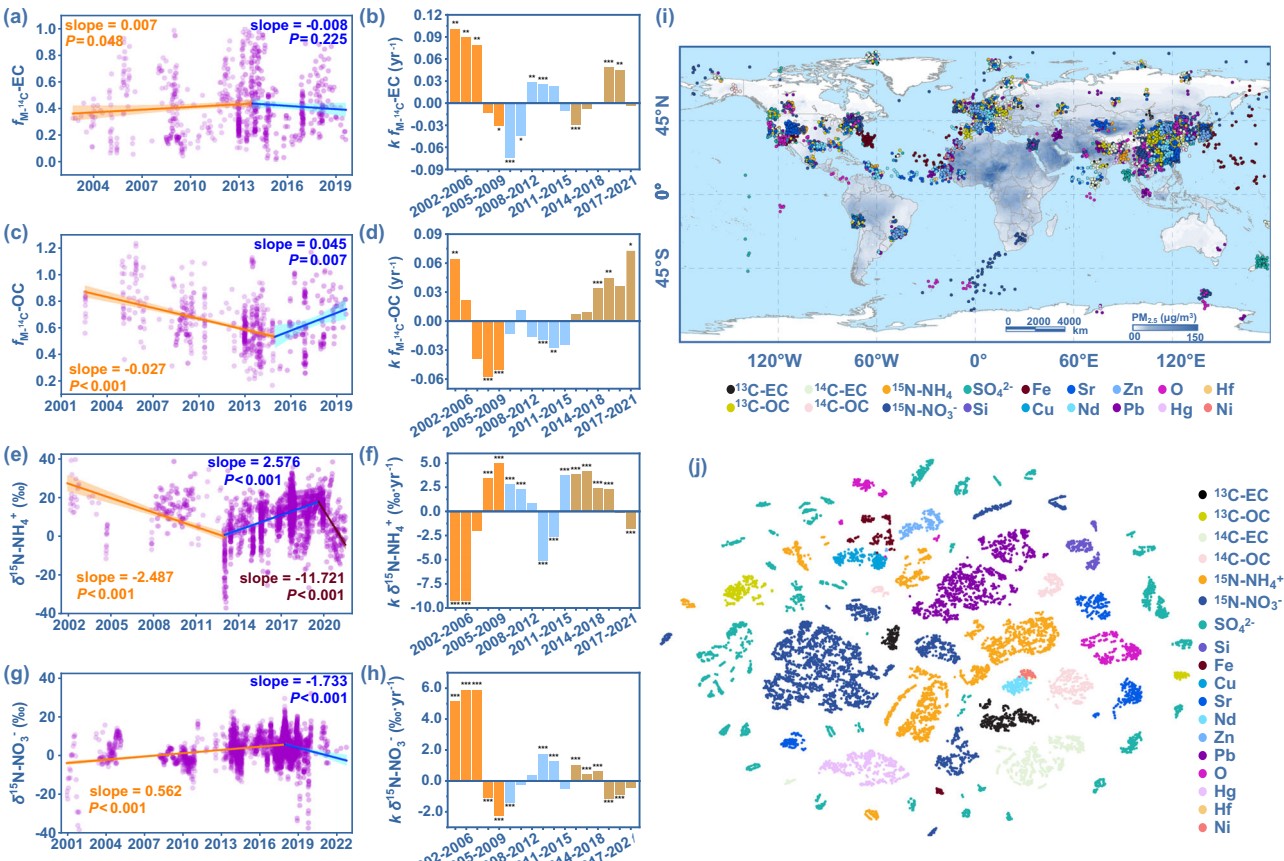

**Fig. 2 | Spatiotemporal variations in global atmospheric particulate matter (PM) isotopic fingerprints. a**, **c**, **e**, **g** Temporal trends of $f_{M-14C}$ for EC (**a**) and OC (**c**) $\delta^{15}N$ for $NH_4^+$ (**e**) and $NO_3^-$ (**g**) were analyzed using formal bootstrap testing procedures. The abbreviations EC, OC, $NH_4^+$, and $NO_3^-$ represent elemental carbon, organic carbon, ammonium, and nitrate, respectively. The different-colored lines in (**a**, **c**, **e**) and **g** represent liner fits for different periods, with the corresponding shading indicating their 95% confidence intervals. **b**, **d**, **f**, **h** Moving subset window analysis of the temporal change trends of $f_{M-14C}$-EC (**b**), $f_{M-14C}$-OC (**d**) $\delta^{15}N$-$NH_4^+$ (**f**) and $\delta^{15}N$-$NO_3^-$ (**h**) over 5-year periods. The analysis step was 1 year. The bars represent isotopic change rates ($k$) every 5 years. Details on the amount of isotopic data in each plot are provided in the Source Data file. ***$P < 0.001$, **$P < 0.01$,

*$P < 0.05$. Statistical robustness was verified using the Theil−Sen estimator (Supplementary Figs. 6 & 7). The color scale of the bars correspond to temporal trends, without specific indication. **i** Global distribution of atmospheric particulate matter isotopic observations and pollution levels. The global atmospheric particulate matter pollution levels are represented by the annual average concentrations of atmospheric fine particulate matter ($PM_{2.5}$) for 2022. Isotopic observations are primarily concentrated in Asia, Europe, and the Americas. The map was generated with Natural Earth, which provides free vector and raster map data available at naturalearthdata.com. **j** The t-distributed stochastic neighbor embedding (t-SNE) map shows global atmospheric particulate matter isotopic affinities within the IDGAR. Most isotopic data form distinct clusters rather than random distributions.

To further characterize the relationships among multi-species isotopic, we performed t-distributed stochastic neighbor embedding (t-SNE) analysis on the isotopic fingerprints from 1890 global PM pollution events. Despite being distributed across various global regions (from 69.37° S to 85.64° N, Fig. 2i) and potentially subject to different interventions at different times, isotopes of the same components tend to cluster into unique groups rather than distribute randomly (Fig. 2j). This highlights the independent indicative capacities of these isotopes and the importance of considering multi-dimensional isotopes in PM source tracing.

**Emission constraint of PM₂.₅ using isotopic big data**
Using the isotopic big data, the Bayesian stable isotope mixing model implemented in the R package (MixSIAR), the mass concentrations of various $PM_{2.5}$ components, we conducted source tracing analyzes for individual $PM_{2.5}$ compounds. By aggregating the emissions of different components, we estimated the contribution of each subdivided source to $PM_{2.5}$. The uncertainty of source apportionment results was derived from MixSIAR model analysis and error propagation formula (see "Methods" for details). Noteworthy, although the isotopic fractionation correction of certain components, such as $NH_4^+$ and $NO_3^-$, during

the conversion process has been accounted for in the MixSIAR model (see "Methods" and Supplementary Note 3 for details), future in-depth studies on the isotopic fractionation mechanisms of specific transformation reactions under real-world environmental conditions will further improve the accuracy of source apportionment results. Furthermore, the robustness of MixSIAR performance was verified using the other Bayesian stable isotope mixing models, i.e., MixSIR and Food Reconstruction Using Isotopic Transferred Signals (FRUITS) (Supplementary Note 4).

Significant variations were observed in the contributions of different sources to $PM_{2.5}$ and its components (Fig. 3). Globally, biomass burning and coal combustion are the main sources of $PM_{2.5}$, and the main component affected by these sources is most likely OC (Fig. 3a). Additionally, biomass burning and vehicle exhaust are the leading contributors to EC in $PM_{2.5}$, accounting for $2.17 \pm 0.09\%$ and $2.23 \pm 0.09\%$ of the total mass of $PM_{2.5}$, respectively, through EC emissions. Natural soil and coal combustion are the major sources of Si in $PM_{2.5}$, with Si emissions from these sources contributing $1.99 \pm 0.08$ % and $1.22 \pm 0.06\%$, respectively, to the overall $PM_{2.5}$ mass. Additionally, some subdivided sources also affect $PM_{2.5}$ pollution levels by emitting specific components. For instance, the combustion of liquid

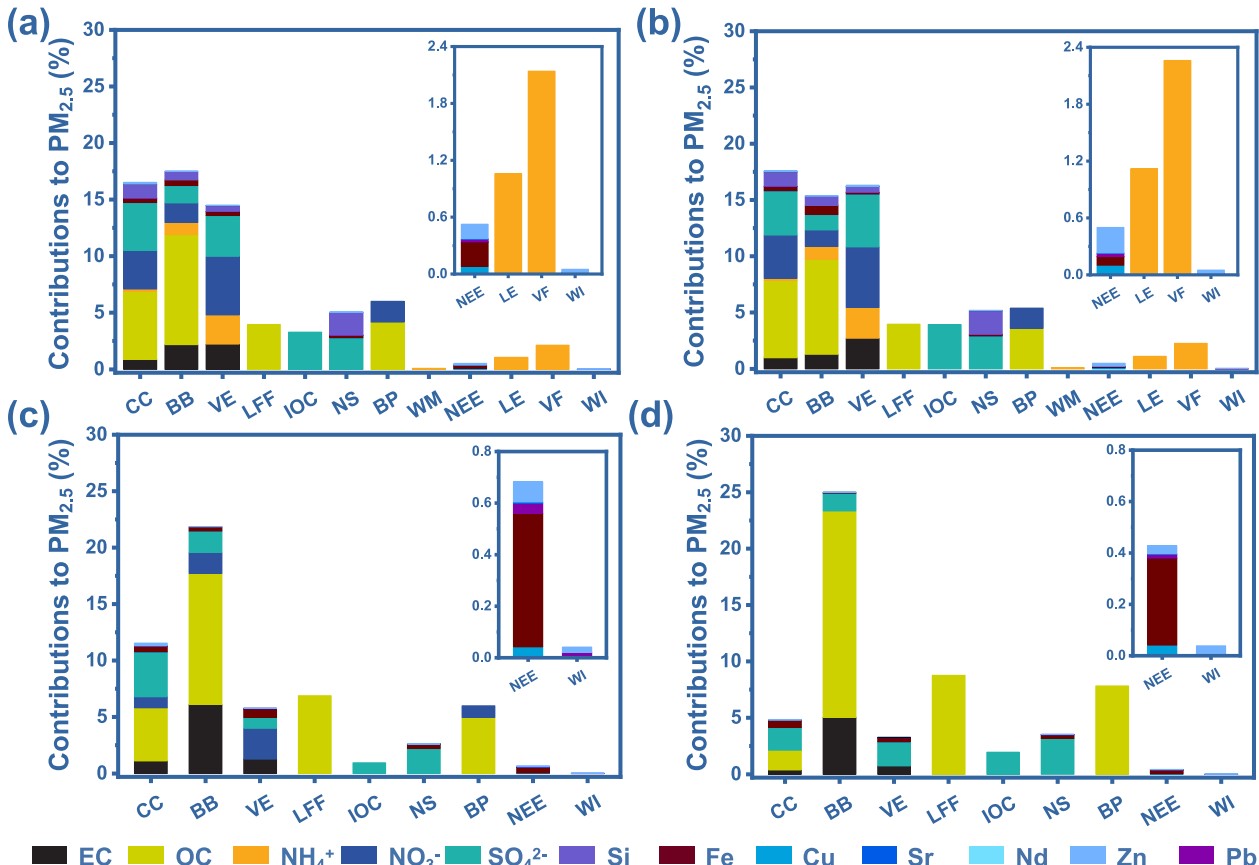

**Fig. 3 | Contributions of individual sources to atmospheric fine particulate matter (PM$_{2.5}$) and its components from 2001–2023. a, b, c, d** The source composition of PM$_{2.5}$ and its components in the worldwide (**a**) Asia (**b**) the Americas (**c**) and Europe (**d**). The labels CC, BB, VE, LFF, IOC, NS, BP, WM, NEE, LE, VF, and WI are abbreviations for coal combustion, biomass burning, vehicle exhausts, liquid fossil fuels combustion, industrial oil combustion, natural soil, biological process, waste materials, non-exhaust emissions, livestock emissions, volatilized fertilizer, and waste incinerator, respectively. Biological processes include plant debris, fungal bacteria, pollen, particulate matter generated by the oxidation of biogenic

volatile organic compound emissions, microbial processes. The uncertainties for these results are provided in the Source Data file. For the specific calculation process, see the "Methods" section. *Note:* in the isotopic source tracing analysis of OC in PM$_{2.5}$, the liquid fossil fuels combustion is classified as a subdivided source, while OC emissions from vehicle exhaust are not double-counted. Due to the absence of the $\delta^{15}$N values for ammonia from biomass burning in the Americas and Europe, the global source apportionment study relied on the $\delta^{15}$N values for ammonia from Asian biomass burning to estimate global values.

fossil fuels and industrial fuel oil primarily impacts PM$_{2.5}$ levels by altering concentrations of OC and sulfates. Non-exhaust emissions are a key source of metal elements in PM$_{2.5}$, especially Fe and Zn, with Fe and Zn emissions contributing 0.26 ± 0.02% and 0.15 ± 0.01% to the total mass of PM$_{2.5}$, respectively. Waste materials, livestock waste, and volatilized fertilizer influence PM$_{2.5}$ pollution levels through the NH$_3$ emission. These refined source apportionment results are essential for the precise implementation of PM$_{2.5}$ source control measures.

Regionally, a closer characterization revealed heterogeneity in the source composition of PM$_{2.5}$ and its components across Asia, the Americas, and Europe. From 2001 to 2023, coal combustion is the dominant source of PM$_{2.5}$ emissions in Asia, whereas biomass burning serves as the primary contributor to PM$_{2.5}$ emissions in Europe and the Americas (Fig. 3b–d). This finding aligns with high coal consumption in Asia[23]. Moreover, the same type of sources, such as biomass burning, show varying contributions to PM$_{2.5}$ components across Asia, the Americas, and Europe. For instance, biomass burning in Asia contributes -1.29 ± 0.82% of PM$_{2.5}$ through EC emissions, which is lower than its contribution in the Americas (6.12 ± 1.70%) and Europe (5.06 ± 4.30%). This result may be attributed to the higher frequency of wildfire emissions in Europe and the Americas[24]. Noteworthy, some subdivided sources show regional differences in PM$_{2.5}$ components and contributions. For instance, industrial oil combustion contributes

substantially more sulfate to PM$_{2.5}$ in Asia than in Europe and the Americas. These differences may be attributed to the industrial structures in different regions.

We further analyzed the temporal changes of PM$_{2.5}$ source composition across Asia, the Americas, and Europe. In Asia, the contribution of coal combustion has gradually decreased since 2013, probably due to interventions targeting OC emissions (Supplementary Figs. 10a–f). Since 2004, the contribution of biomass burning to PM$_{2.5}$ has remained stable. Yet, vehicle exhaust emissions have improved significantly since 2018, likely due to the rapid development of new energy vehicles[25]. Notably, contributions from some subdivided sources—such as liquid fossil fuel combustion, industrial fuel oil combustion, and waste incineration—declined significantly after 2013–2015 but have stabilized since 2016. These results might reflect the effective implementation of targeted regional emission reduction measures. In the Americas and Europe, the contribution of most sources, e.g., coal combustion, vehicle exhaust, and livestock waste, to PM$_{2.5}$ has shown a gradual downward trend (Supplementary Fig. 10g–l). In contrast, contributions from biomass burning and liquid fossil fuel combustion have either remained stable or increased, suggesting these sources necessitate greater attention and further intervention. Notably, current source-tracing analysis mainly relies on the critical components of PM$_{2.5}$, where isotope fingerprints can be

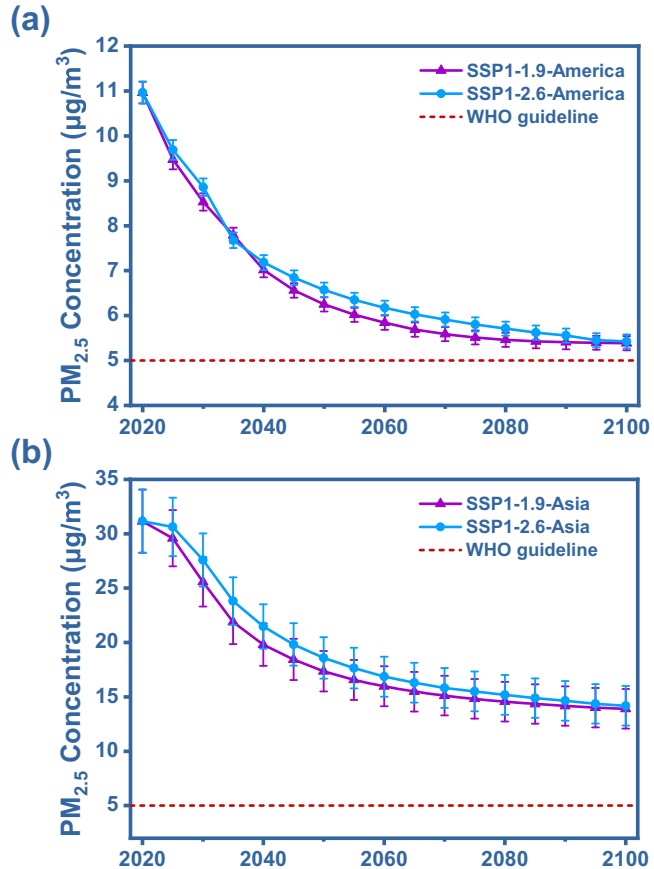

**Fig. 4 | Projected atmospheric fine particulate matter (PM$_{2.5}$) pollution trends until 2100 for the Americas and Asia.** Specifically, (**a**) the Americas, (**b**) Asia. Simulations were conducted with the global change assessment model (GCAM), IDGAR database, and the stable isotope analysis in R model (SIAR). The scenarios, SSP1-1.9 and SSP1-2.6, combine Shared Socioeconomic Pathways (SSP) with Representative Concentration Pathways (RCP), which correspond to future radiative forcing and their climate impacts. Specifically, SSP1-1.9 and SSP1-2.6 aim to limit global warming to 1.5 °C and 2 °C, respectively. Error bars represent mean ± s.d. (*n* = 3). For more details about the scenario settings in the PM pollution simulations, see the "Methods" section. These projected results by 2100 are close to the WHO guideline limit of 5 μg/m$^3$, but have not yet been achieved.

obtained. As more component isotopic fingerprints become available, e.g., NH$_4^+$ in the Americas and Europe, the accuracy and precision of source tracing will further improve.

**Effect of future PM$_{2.5}$ interventions considering climate change**
In addition to current interventions, global climate actions aimed at controlling warming will lead to systemic change in PM$_{2.5}$ emissions, e.g., fossil fuels[26–28]. To better plan future PM$_{2.5}$ interventions, we projected the trends of PM$_{2.5}$ pollution levels (Fig. 4) and isotopic fingerprints (Supplementary Fig. 11) in the Americas and Asia in climate mitigation scenarios to the year 2100. This simulation was conducted by integrating the IDGAR database, the global change assessment model (GCAM), and the weather research and forecasting model coupled with chemistry analysis (WRF-Chem). For further details see "Method" section. In both the Shared Socioeconomic Pathways SSP1–1.9 (1.5 °C temperature control target) and SSP1–2.6 (2 °C goal temperature control target) scenarios, PM$_{2.5}$ pollution levels in Asia and the Americas are projected to continuously decline. By 2100, with anthropogenic sources expected to have achieve net zero emissions (Supplementary Fig. 12), the PM$_{2.5}$ concentrations in the Americas in the SSP1–1.9 and SSP1–2.6 scenarios are projected to

decrease to 5.38 ± 0.16 μg/m$^3$ and 5.43 ± 0.15 μg/m$^3$, respectively. These results are close to the WHO guideline limit of 5 μg/m$^3$ but do not quite achieve it. In Asia, the projected PM$_{2.5}$ concentrations for 2100 are 13.9 ± 1.82 μg/m$^3$ and 14.2 ± 1.82 μg/m$^3$ in the SSP1-2.6 and SSP1-1.9 scenarios, respectively. These PM$_{2.5}$ levels are comparable to current levels in the Americas and fall short of the WHO recommended limits. Compared to that in the SSP1–2.6 scenario, the PM$_{2.5}$ concentration decreases more in the SSP1–1.9 scenario, but the difference is relatively small. Hence, we propose that, in addition to implementing stricter climate targets, designing targeted interventions remains crucial for controlling PM$_{2.5}$ pollution, particularly for Asia.

## Discussion
Addressing the current limitations of isotope-based PM$_{2.5}$ driver tracing, which primarily focused on local-scale, short-term pollution events involving single component, we constructed the global atmospheric isotopic database, IDGAR. Through global long-term temporal dynamic analyzes of isotopic big data, we identified the main sources and corresponding interventions for different PM$_{2.5}$ species over the past two decades. Moreover, we achieved the source constraints on PM and its components based on IDGAR's multi-species isotopic fingerprints. However, projections indicate that WHO's recommended PM$_{2.5}$ limit of 5 μg/m$^3$ will not be met by 2100 under the current climate targets, need additional effective interventions, particularly in natural emissions.

Blockchain technology is critical for the accuracy and sustainability of IDGAR isotopic big data. It ensures each isotopic data point is unique, immutable, and traceable, thereby enhancing the accuracy of PM source tracing and intervention assessments. Given that isotopic analysis requires a high level of analytical capability and considerable time, the amount of isotopic data from a single laboratory is very limited, necessitating global trusted collaboration. Blockchain's tamper-proof and traceability features make IDGAR a trustworthy platform, encouraging researchers to enrich and utilize its isotopic data. Moreover, unlike traditional administrator-based databases, blockchain-based IDGAR implements a decentralized data management model, ensuring equal data management rights for all users.

Retrospective analysis of isotopic big data reveals remarkable intervention effects on C4 plants burning, coal combustion, or vehicle exhaust, particularly for EC, OC, NO$_3^-$, NH$_4^+$, SO$_4^{2-}$, Pb-bearing PM. Our study have identified the main sources of most curated element species but not all, e.g., Fe-bearing PM, Zn-bearing PM, which require targeted interventions. In contrast to previous measure-specific intervention assessments based on bottom-up emission inventories[29], our study provides a complementary top-down, component-level assessment, mitigating the impact of the nonlinear relationship between emissions and PM concentrations. Notably, hardly any PM species has been effectively intervened continuously in the past 20 years. This probably because the main sources of the target components have changed, but the intervention designs have not yet to respond. The asynchronous changes in the main sources and intervention effectiveness suggest that future intervention designs should be dynamically adjusted for different PM species. An ideal approach would be to develop an isotopic-guided intelligent system for designing interventions for different PM species within the IDGAR framework.

Source tracing utilizing isotopic big data reveals the specific emissions of critical PM$_{2.5}$ components, which is essential for implementing precise source control. While the temporal changes in source composition highlight notable intervention effects on coal combustion, resulting in reductions of OC-bearing PM emissions in Asia, the Americas, and Europe. Additionally, we further reveal the increasing impact of biomass burning and liquid fossil fuels combustion on PM$_{2.5}$ levels in these three regions. Projections of PM$_{2.5}$ trends under

scenarios aimed at achieving climate goals of 1.5 °C and 2 °C also underscore the importance of implementing interventions on biomass burning to meet the WHO's annual $PM_{2.5}$ guideline of 5 µg/m³. This is consistent with prior projections indicating that climate-driven natural emissions, e.g., wildfire smoke, will worsen $PM_{2.5}$ exposure[30,31].

In the Americas, satellite observations and global climate model projections show that biomass burning, e.g., wildfire smoke, will increasingly impact PM pollution[24], intensifying with ongoing climate warming[32,33]. Therefore, developing interventions targeting wildfire may be effective for reaching WHO recommended threshold in the Americas. However, managing wildfires requires specialized interventions distinct from those employed for regional anthropogenic emission, which are not adequately covered by current air pollution regulations[34]. In Asia, the projected lowest $PM_{2.5}$ levels will still be 13.9 ± 1.82 µg/m³ in 2100, comparable to current levels in the Americas. In addition to wildfires, the notable increase in biomass burning emissions in Asia is largely due to extensive agricultural activities, particularly in rural South Asia[8,22,35–37]. Moreover, soil-related dust is another important driver of PM pollution, especially in Central and East Asia[38]. Therefore, simultaneous and intensive interventions targeting agricultural biomass burning, wildfires, and dust are necessary in Asia to achieve WHO guidelines.

In summary, we have launched the blockchain-based isotopic database for global atmospheric research. This database provides region-specific source-end isotopic values for PM tracing. Our analysis of long-term trends and change rates of isotopic big data reveals varying main sources and asynchronous interventions for different PM species. Predictions under climate mitigation scenarios indicate that controlling natural sources, e.g., biomass burning, is essential to meet WHO guidelines. Isotopic big data are critical for atmospheric research, extending beyond PM and emission tracing. The data can be used to quantify PM transport ranges and estimate deposition fluxes. Moreover, integrating isotopic big data with public health or policy factor datasets could contribute to source-specific health effect assessments and intelligent policy making.

## Methods
### Data curation
The database developed in this study includes 34,815 non-redundant isotopic data points related to PM species and corresponding emissions, as recorded up to March 31, 2024. This database includes isotopic fingerprints for compounds and elements such as OC, EC, nitrate, ammonium, and sulfate, Si, Fe, Ni, Cu, Zn, Sr, Nd, Hf, Pb, and Hg; different types of PM, e.g., $PM_{2.5}$, $PM_{10}$, and total suspended particulate matter (TSP); different PM sources, such as e.g., biomass burning, coal combustion, vehicle exhausts, waste materials, volatilized fertilizer, livestock emissions, microbial processes, industrial oil combustion, natural soil; and additional details such as specific components, sampling methods, locations, time periods, and analytical methods/measurement techniques. For detailed information, please refer to Supplementary Table 3–5. Briefly, the sampling of $NH_3$ from sources is categorized into active and passive sampling. For source apportionment applications, data correction for passive sampling is performed by adding 15.4‰, following commonly used methods in the literature[39]. Source emissions of $NO_X$ are collected through active sampling using the standard solution absorption method[40]. Similarly, almost source emissions of $SO_2$ are also collected via active sampling with solution absorption. Since only one study used passive sampling for $SO_2$, we excluded its reported values from S source tracing to ensure data standardization and the reliability of the research findings[41]. Aside from $NH_3$, $NO_x$, and $SO_2$, other PM components (such as OC, EC, Si, and metals) included in the database are not sampled individually but are collected through active sampling of source emissions, followed by the analysis of the specific isotopic fingerprint characteristics of the different components within the particulate

matter. Additionally, different components may be analyzed using various methods. For instance, the nitrogen isotope measurement is conducted using both chemical conversion and bacterial conversion methods; for Strontium (Sr) isotopic measurement might be performed using thermal-ionization mass spectrometry (TIMS) or multi-collector inductively coupled plasma mass spectrometry (MC-ICP-MS), both of which undergo strict quality control. The isotopic results from different analytical methods are reported with reference to international standard reference materials, ensuring consistency and standardization. Regarding different standard reference materials, such as Zn, the Zn isotopic composition is typically expressed in per mil (‰) relative to the "zero point" of the isotopic standard reference material, IRMM-3702. Notably, some publications report the Zn isotopic composition relative to JMC 3-0749 L Lyon (another reference material). The Zn isotopic composition of JMC 3-0749 L Lyon is +0.32‰ relative to IRMM-3702. For clarity, the Zn isotopic signatures presented here have been converted to be relative to IRMM-3702. Notably, given the importance of data standardization and the emergence of more precise cross-validation methods, the data in this database needs to be standardized and updated.

The data were extracted from 804 publications (including peer-reviewed papers, conference papers.) using Web Plot Digitizer (version 4.2, San Francisco, California, USA). The publications were identified through manual searches on Web of Science (http://isiknowledge.com), Google Scholar (http://scholar.google.com), and Baidu Scholar (http://xueshu.baidu.com). The specific keywords, number of samples, type of data, and specific reference are provided in Supplementary Table 3 and "References" in Supplementary Information. These information can also be found in IDGAR database (http://idgar.org/). ArcGIS Pro (ESRI; Environmental Systems Research Institute, Redlands, CA, USA) was used to visualize the specific geographic locations of these isotopic data. The base map was downloaded from https://www.naturalearthdata.com/downloads/110m-cultural-vectors/.

In addition to radiocarbon (¹⁴C) and Pb, the isotopic fingerprints of other elements are reported as delta values (δ) relative to an isotopic standard reference material (SRM) using the following formula:

$$\delta^x E = \left( \frac{(x_E/y_E)_{sample}}{(x_E/y_E)_{standard}} - 1 \right) \times 1000 \qquad (1)$$

where E represents a selected element and $x$ and $y$ describe the mass numbers of the isotopes of element E.

Notably, the isotope analysis methods used for an element may vary among publications. This expression, relative to an SRM with a "zero point", makes it feasible to perform comprehensive analyzes based on multiple isotopes. The specific SRMs (Supplementary Table 3) and analytical accuracy of these isotopic analyzes are collected in the developed database.

Isotopic fingerprints for radiocarbon (¹⁴C) are reported as the fraction of modern carbon ($f_{M\text{-}14C}$) as follows:

$$f_M = \frac{(14_C/12_C)_{sample}}{(14_C/12_C)_{standard}} \qquad (2)$$

where the standard reference material is the oxalic acid II 4990 C from the National Institute of Standards and Technology (NIST), USA[21,42].

Due to the lack of a unified standard reference material, Pb isotopic signatures are reported as ²⁰⁶Pb/²⁰⁴Pb, ²⁰⁷Pb/²⁰⁴Pb, ²⁰⁸Pb/²⁰⁴Pb, ²⁰⁷Pb/²⁰⁶Pb, ²⁰⁸Pb/²⁰⁶Pb, and ²⁰⁸Pb/²⁰⁷Pb.

Additionally, the data of PM concentrations in 2021 were sourced from the public dataset of ground-level $PM_{2.5}$, measured at a resolution of 0.1° × 0.1° from 1998 to 2022 (V5.GL.04), calculated by the Atmospheric Composition Analysis Group[43].

## Portal implementation of the blockchain-based isotopic database

A web portal for the IDGAR including PM data and the corresponding emissions was established using blockchain technology. The IDGAR website was developed using the Spring Boot framework with the MVC architecture pattern, and MariaDB was utilized for data storage. The website is hosted on Amazon Web Services (AWS) and is publicly available at http://idgar.org/. Moreover, blockchain technology is used to secure the database against data tampering and to support operations across multiple nodes. The SHA256 hashing function is used to process data: user registration details are hashed, generating a 64-character hexadecimal string. This hash string is then stored across various database nodes for security. Each user's information hash string forms a block, and the blocks are connected in chronological order to create a blockchain. The IDGAR includes a data validation procedure, in which the hash strings are continuously recalculated in real time (every 5 minutes). If any discrepancies in the hash strings are detected, we can precisely identify the affected block and tracing the origins for non-compliance.

Newly uploaded data undergo real-time hash string validation and are rigorously compared with existing strings in the database. If a conflict occurs, the system automatically assigns incremental version numbers to subsequent file versions. The irreversible nature of blockchain prohibits direct editing of uploaded data in the IDGAR. If a user makes an honest error and needs to correct the uploaded content, they can submit the corrected data with a clear statement, automatically generating incremented version numbers for the file. The old data remain intact in the blockchain at all times.

Moreover, an early warning system activates promptly if a node is attacked, transferring all the data once the network virus is eradicated. This decentralized management fosters equitable data sharing. Notably, unpublished data are not decentralized but rather controlled by the uploader using password-based key derivation encryption. Users who receive the shared key from the uploader can access it. With the development of the IDGAR, unpublished data can be uploaded with clear statements to foster academic collaboration.

## Statistical analyzes

One-way analyzes and Pearson correlation analyzes were carried out by using SigmaPlot version 12.5 (Systat Software, Inc., Chicago, IL, USA) and the Origin 2016 statistical package (OriginLab Corporation, USA). The remaining statistical analyzes were conducted in R version 4.1.3 using the "stats" package, unless stated otherwise. Briefly, the temporal trends of the PM isotopic fingerprints were analyzed via bootstrap procedures. The method of formal testing was used to identify the breakpoints from all years. $P$ values < 0.05 were considered statistically significant. The Theil-Sen estimator was employed to test the robustness of the results. Moreover, a one-dimensional moving subset window analysis was employed to estimate the temporal trends in the PM isotopic composition from 2001 to 2022, including dependencies on longitude and latitude for source isotopic changes. For each isotope, the data were segmented into several subsets based on sampling times or locations, with each subset encompassing several consecutive years, longitudes, or latitudes. The initial subset was established as the starting point. Subsequent subsets were formed by excluding data from the earliest year or site of the previous subset and incorporating an additional step. The slope and $P$ value of the linear regression between the isotopic composition and year or site information in each subset indicate the rate ($k$) of change of the isotopic composition and its significance over the respective period or geographic range.

To visualize the relationships among all the data, all the isotopic data, along with the PM type, concentration, sampling time, latitude, and longitude, are presented in a 2D figure. Briefly, all of the data are uniformly transformed into corresponding 64-dimensional vectors through a shallow neural network based on a given corpus. This process was carried out in Python version 4.1 with the "gensim" package. Then, dimensionality reduction was performed using t-distributed stochastic neighbor embedding (t-SNE), a machine learning technique for dimensionality reduction of multivariate statistical data[26], after preprocessing with principal component analysis (PCA), was carried out in R version 4.1.3 with the "tsne" package.

## Source apportionment of PM2.5 species

The relative proportions of the selected species (e.g., compounds, metals) in PM from diverse emissions are quantitatively estimated by the Bayesian MixSIAR model. This model was performed in R version 4.1.3 with the "MixSIAR" package. Considering increasing the data volume and the number of simulations can effectively improve the reliability of the analysis, we conducted source tracing analysis of PM components using IDGAR isotopic big data rather than relying on a few scattered data points. Moreover, we set the number of iterations to 1,000,000 with a burn-in of 500,000 in our MixSIAR simulation. Furthermore, the Isospace plots and Markov chain Monte Carlo diagnostic tests (including Gelman-Rubin and Geweke) have been incorporated into the MixSIAR package to ensure the results are as accurate as possible. The MixSIAR method can identify the most reasonable source composition of a mixture based on isotope mass balance[5].

Moreover, the fractionation effect of some isotopes (e.g., $NH_4^+$) and the variabilities in isotopic fingerprints of measurements and sampling can be incorporated into this model (see Supplementary Note 3 for details)[6]. For OC, we first determined the contributions of fossil and non-fossil fuel combustion using $f_{M14C}$-OC, followed by a detailed analysis of specific fossil and non-fossil fuel contributions using the isotopic chemical mass balance of $\delta^{13}C$[44]. The overall calculation principle of source tracing can be expressed as follows:

$$\delta^x E_{PM} = \sum (\delta^x E_i \times f_i) \tag{3}$$

where $\delta^x E_{PM}$ represents the isotopic fingerprint of the PM, $\delta^x E_i$ is the isotopic composition of potential sources. The $f_i$ values describe the relative contributions of the emissions to the selected species in the PM.

Afterward, using the $f_i$ value multiply the mass concentration (%) of the selected compound or element (e.g., EC) in PM, we can estimate the weight of this component emitted from each source to PM2.5. The detailed mass concentrations of the curated species in PM2.5 are provided in Supplementary Fig. 2a. The calculation processes can be expressed as follows:

$$F_i = C_i \times f_i \tag{4}$$

where F represents the relative contributions of sources to PM2.5 by emitting the selected component. The f values are source contributions to the selected component. C (%) reflects the mass concentrations of the selected components in the PM2.5.

The uncertainty in source apportionment was derived from the MixSIAR model analysis and the error propagation formula. The detailed calculation process is as follows:

$$\sigma_{Ssum} = \sqrt{\sum_{i=1}^{n} \left[ (X_E \times C_{ES}) \times \sqrt{\left(\frac{\sigma_{X_E}}{X_E}\right)^2 + \left(\frac{\sigma_{C_{ES}}}{C_{ES}}\right)^2} \right]} \tag{5}$$

where $\sigma_{Ssum}$ represents the standard deviation of the total contributions from source 'S' to PM2.5. $X_E$ denotes the mass concentration of component 'E' in PM2.5, and $C_{ES}$ represents the relative contribution of source 'S' to component 'E' in PM2.5, as calculated by MixSIAR. Additionally, $\sigma X_E$ indicates the uncertainty in the mass concentration of component 'E' in PM2.5, while $\sigma C_{ES}$ reflects the uncertainty in the $C_{ES}$ values derived from MixSIAR.

To further validate this method, we used $f_{M-14C}$ to evaluate the $\delta^{13}C$-based source tracing results of EC in PM$_{2.5}$. Specifically, we calculated the $f_{M-14C}$ isotopic compositions of EC in PM$_{2.5}$ based on the $f_{M-14C}$ of primary sources and $\delta^{13}C$-based relative contributions, and compared them with the observed values. As shown in Supplementary Fig. 13, the calculated $f_{M-14C}$ isotopic compositions of EC were consistent with the observed values. The largest net difference between the calculated values and observed data was 0.05, which was smaller than the observed uncertainty of 0.07[45]. This supported the robustness of the $\delta^{13}C$-based source tracing results. Furthermore, the comparison of model performance among MixSIAR, MixSIR, and FRUITS confirms the robustness of the source apportionment results in this study (see Supplementary Note 4 for details).

## PM$_{2.5}$ pollution simulation and future scenario setting

The framework for PM$_{2.5}$ pollution simulation in the Americas and Asia comprises two sub-sections, i.e., emission projection and air quality modeling. The future emission changes of diverse sectors under climate mitigation scenarios towards 2100 were performed with the global change assessment model (GCAM). Specifically, we selected two scenarios, SSP1–1.9 and SSP1–2.6, which integrate different Shared Socioeconomic Pathways (SSPs) with climate projections based on Representative Concentration Pathways (RCPs). SSPs show the challenges of climate change mitigation and adaptation, capturing key socioeconomic factors. RCPs reflect the levels of ambition in climate policies. The scenarios SSP1–1.9 (SSP1 + RCP1.9) and SSP1–2.6 (SSP1 + RCP2.6) align with global climate goals of limiting warming to 1.5 °C and 2 °C, respectively. These scenarios represent the most likely patterns of global development in the future. Using updated emission data, we further estimated the trends in PM$_{2.5}$ pollution and their isotopic composition through the weather research and forecasting model coupled with chemistry analysis (WRF-Chem) and the SIAR model. It is worth noting that, due to noticeable uncertainty in emission inventories, the predicted PM$_{2.5}$ pollution levels usually require optimization and correction. Isotopic-based source tracing results can refine the inventory from a top-down perspective, enhancing the accuracy of PM$_{2.5}$ predictions (Supplementary Fig. 14).

## Data availability

The raw data used in this study are available in the IDGAR database [http://idgar.org/][17]. Source data are provided with this paper.

## Code availability

The code used in this study are also available in the [https://doi.org/10.5281/zenodo.15064820][46].

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

## Acknowledgements

This work was financially supported by the National Natural Science Foundation of China (Grants. 22222610, 22376202, 22193051 to D.L., 42107124 to J. Z., 42477099 to Z. Z.), the Strategic Priority Research Program of the Chinese Academy of Sciences (Grant No. XDB0750100 to D.L.), the National Key R&D Program of China (2023YFF0614200 and 2023YFC3708301 to D.L.), Chinese Academy of Sciences Project for Young Scientists in Basic Research (YSBR-086 to D.L.), the Taishan Scholars Program of Shandong Province (tsqn202306084 to Z. Z.).

## Author contributions

D.L. conceived and designed the research. Y.H., Y.W., X.Li., J.Z., Z.Z., and D.L. performed data curation and analyzes. Y.L., C.X., X.Liu., Q.L., G.M., and G.J. participated in the discussion. Z.Z., W.N., J.L., and J.Z. helped develop the mathematical models. D.L., Y.H., Z.Z., J.Z., and J.L. wrote the paper.

## Competing interests

The authors declare no competing interests.
