## [Transparent Peer Review file · Nature Communications]

Blockchain-based isotopic big data-driven tracing of global PM sources and interventions

Corresponding Author: Dr Jingwei Zhang

Version 0:

Reviewer comments:

Reviewer #1

(Remarks to the Author)

This study collected a large historical PM isotopic dataset for source apportionment of PM on the global scale, and the PM trends in climate mitigation scenarios were projected. The dataset is open sourced and blockchain based. The sources of PM are distinct and dynamically changed over time. The paper and the dataset are potentially important. However, the following concerns should be addressed before consideration of publication.

Main comments:

First, regarding the novelty, it is well-known that isotopic method can provide accurate tracing of PM emission sources. Therefore, technically, source apportionment of PM using isotopic technique is not entirely new, although this study has largely expanded the study regime. I suggest that the authors should emphasize the spatial and temporal characteristics of PM sources, to enhance the novelty of this study.

Second, the architecture of the database has been greatly emphasized as a decentralized blockchain based. However, the dataset itself does not really relate to any scientific findings. It could be a different data paper for journal like 'Scientific Data' rather than being emphasized in this paper. The data are presented as Figure 1. It is a bit strange to show the architecture of database rather than the main scientific findings as the first figure.

Specific comments:

1. The application of blockchain technology to isotopic data management is interesting, but more details should be provided.
2. What is the size range of the PM analyzed in this paper? PM1 or PM2.5?
3. The description of future simulation using SSP data and GCAM is unclear. It is unclear how the source data provided in this paper can be used for future PM simulation.

(Remarks on code availability)

Reviewer #2

(Remarks to the Author)

In recent years, a variety of data has been conducted from research on aerosols using stable isotope ratios, and this research is generating extremely valuable data. This research is a new challenge using a groundbreaking method that has never been used before, using a huge amount of data: 18,760 non-redundant global isotopic observations, including 7,153 sources and 11,607 PM isotopic data published between 1967 and 2024. However, there are several fundamental problems with this paper, and I think that this paper should not be accepted by Nature Communications. The reasons are described below.

Although analysis was done using isotopes of 14 elements (carbon ($\delta^{14}\text{C}$ for radioactive; $\delta^{13}\text{C}$ for stable), nitrogen ($\delta^{15}\text{N}$), oxygen ($\delta^{18}\text{O}$), silicon ($\delta^{30}\text{Si}$), sulfur ($\delta^{34}\text{S}$), iron ($\delta^{56}\text{Fe}$), nickel ($\delta^{60}\text{Ni}$), copper ($\delta^{65}\text{Cu}$), zinc ($\delta^{66}\text{Zn}$), strontium

($\delta^{87}\text{Sr}$), neodymium ($\delta^{144}\text{Nd}$), hafnium ($\delta^{177}\text{Hf}$), lead ($^{207}\text{Pb}/^{206}\text{Pb}$), and mercury ($\delta^{202}\text{Hg}$)), some elements are in the form of compounds in aerosols. For example, carbon is composed of inorganic carbon and organic carbon, and nitrogen is mainly composed of different compounds such as nitrate and ammonium salts. In the study of stable isotope aerosols, it was initially impossible to analyze the stable isotope ratios in compounds due to analytical problems, and bulk isotope analysis was the mainstream, but in recent years, isotope analysis of each compound has become the mainstream. In addition, without analyzing the stable isotope ratios of each compound, it is impossible to estimate the origin, and there was a problem in interpreting these data. In this paper, the results are interpreted as the results of the stable isotope ratios of the bulk, but the research is meaningless. If the research were to be conducted based on this idea, it would seem that one approach would be to use only heavy metal isotopes, rather than using isotopes of light elements that form compounds.

In addition, the sources of each individual isotope compound are divided into four (biomass burning emissions, coal combustion and industrial emissions, soil-related emissions, and vehicle emissions). However, since there are different sources for each compound, there is a problem with combining these, and the analysis may become meaningless. In addition, the analysis was performed using the isotope mixing model SIAR, but the validity of these is also very uncertain.

In addition, although the topic of global warming is described as the purpose of the research, it was unclear because there was no description of how it relates to aerosol research.

(Remarks on code availability)

I just checked the website.

Version 1:

Reviewer comments:

Reviewer #2

(Remarks to the Author)

I highly appreciate the innovative approach taken in this study, particularly the transition from bulk isotopic ratios to compound specific isotopic analysis. This is a significant advancement that enhances our ability to understand the sources of PM_{2.5} and provides a valuable framework for future research. However, I believe that several critical issues remain unresolved. Addressing these concerns will be essential to ensure that the paper meets the standards of Nature Communications.

First, for components such as ammonium and nitrate ions, it is important to consider isotopic fractionation during the conversion of precursor gases (NH_3 and NO_x) into aerosols. These processes can lead to significant isotopic shifts, which must be accounted for when interpreting the sources in a real-world environment. The current manuscript does not appear to discuss this aspect. I suggest focusing on components where isotopic fractionation is negligible (e.g., elemental carbon or heavy metals) or framing the study as a preliminary effort to compile source data rather than attempting to draw definitive conclusions about source apportionment.

Second, I found it unclear how source contributions were calculated in cases where source-specific data are missing. For example, global $\delta^{15}\text{N}$ values for ammonia from biomass burning are not yet well-established. Without these critical data, the reliability of the source apportionment results is questionable. This point requires clarification or additional discussion to justify the methodology.

Third, the study relies exclusively on the SIAR model for source apportionment. While this is a well-established approach, there are other models available, such as MixSIR or FRUITS, which may offer complementary insights. Have the authors considered comparing these models, or at least discussing their relative strengths and limitations? Additionally, the paper does not provide a clear discussion of the uncertainties in the calculated source contributions. Including an analysis of uncertainties would significantly strengthen the robustness of the results.

In conclusion, this paper makes an important contribution to the field by introducing a framework for component-level isotopic analysis. However, resolving the above issues will be necessary to enhance the scientific rigor and clarity of the study. Addressing these points would also help the paper better align with the expectations of a journal like Nature Communications.

(Remarks on code availability)

Version 2:

Reviewer comments:

Reviewer #2

(Remarks to the Author)

Review Comments

I highly appreciate the authors' efforts in considering isotopic fractionation during secondary formation and their attempts to explore multiple isotopic mixing models. These aspects significantly enhance the scientific rigor of the study and should be recognized.

However, several concerns remain unresolved. Unless these issues are properly addressed, I believe acceptance in Nature Communications will be challenging.

Major Consideration: Reliability of Source Data

I have significant concerns regarding the reliability of the source data. For example, the nitrogen stable isotope ratios ($\delta^{15}\text{N}$) of ammonia gas varies significantly depending on the sampling method. It has been demonstrated that passive samplers and active samplers can yield $\delta^{15}\text{N}$ differences as large as 15%, as shown by Pan et al. (2020) and Kawashima et al. (2021). Such methodological discrepancies must be carefully considered when integrating source data from multiple studies. Moreover, similar issues may exist for other source data beyond ammonia. This discrepancy is critical because it directly affects the reliability of source apportionment models. However, the manuscript does not indicate whether such methodological differences were considered when integrating source data from multiple studies. Furthermore, similar concerns apply not only to ammonia but also to other source data included in the database. If the same level of scrutiny is not applied across all source categories, the resulting database may lack scientific rigor and lead to misleading conclusions. A comprehensive assessment of data reliability is essential to ensure that the isotopic source apportionment framework remains valid.

To improve the robustness of the study, I strongly recommend that the authors explicitly clarify in manuscript or database:

1. What sampling and analytical methods were used for all source data? – Were the reported isotope values derived from a consistent approach, or do they include data obtained under different methodologies without correction?
2. Which specific sources were included in the dataset? – A table summarizing the locations, time periods, and measurement techniques for each source type (not just ammonia but also elemental carbon, sulfate, metals, etc.) would improve transparency.
3. How methodological bias was addressed? – If different studies employed different sampling or analytical techniques, were correction factors applied? If not, how do the authors justify integrating isotopic data that may not be directly comparable?"

Further Consideration: The Need for a Correctable Database

Given the well-documented impact of sampling and analytical techniques on isotope ratios (Pan et al., 2020; Kawashima et al., 2021), I also suggest that the authors discuss the potential for updating and refining the database as new, more accurate source data become available. A rigid, static database that does not allow for corrections in light of new findings may introduce long-term issues in source apportionment. The authors should consider implementing a mechanism for ongoing validation and revision of the database to ensure its long-term scientific reliability.

Pan, Y., Gu, M., Song, L., Tian, S., Wu, D., Walters, W. W., . . . Wang, Y. (2020). Systematic low bias of passive samplers in characterizing nitrogen isotopic composition of atmospheric ammonia. *Atmospheric Research*, 243, 105018. <https://doi.org/10.1016/j.atmosres.2020.105018>.

Kawashima, H., Ogata, R., & Gunji, T. (2021). Laboratory-based validation of a passive sampler for determination of the nitrogen stable isotope ratio of ammonia gas. *Atmospheric Environment*, 245, 118009. <https://doi.org/10.1016/j.atmosenv.2020.118009>.

Response to Reviewers' Comments

We really appreciate the efforts that you and anonymous reviewers have made to improve the quality of our manuscript titled “Blockchain-based isotopic big data-driven tracing of global PM sources and interventions” (NCOMMS-24-42155-T). We have now made substantial revisions to the manuscript according to the reviewers' comments. Each point raised by the reviewers has been carefully considered and replied beneath. The changes in the manuscript and Supplement are marked in blue font. In this response letter, the comments from reviewers are in blue italic type and author's responses are in black font.

Response to Reviewer #1:

This study collected a large historical PM isotopic dataset for source apportionment of PM on the global scale, and the PM trends in climate mitigation scenarios were projected. The dataset is open sourced and blockchain based. The sources of PM are distinct and dynamically changed over time. The paper and the dataset are potentially important. However, the following concerns should be addressed before consideration of publication.

Main comments:

Question: *First, regarding the novelty, it is well-known that isotopic method can provide accurate tracing of PM emission sources. Therefore, technically, source apportionment of PM using isotopic technique is not entirely new, although this study has largely expanded the study regime. I suggest that the authors should emphasize the spatial and temporal characteristics of PM sources, to enhance the novelty of this study.*

Answer: Thanks for your valuable suggestion. The spatiotemporal characteristics of PM sources and interventions are the focus of this study.

- 1) The isotopic big data provides an unprecedented opportunity for long-term analyses of global PM isotopic fingerprints, enabling the identification of shifts and intervention effectiveness on main sources of PM compounds. For instance, the global f_{M-14C} of EC in PM increased significantly before 2014, then declined (**Reply Fig. 1a**), indicating that their main emissions shift from non-fossil fuel source (i.e., biomass burning, $f_{M-14C} = 1$) to fossil fuel combustion (coal or oil, $f_{M-14C} = 0$) in 2014. By analyzing the $\delta^{13}C$ -EC trend and source isotopic values, we further

identified whether biomass burning before 2014 was dominated by C4 or C3 plants (**Reply Fig. 1e**). Specifically, the rise in $\delta^{13}\text{C-EC}$ during 2001–2008 and 2012–2014 suggests C4 plants burning ($\delta^{13}\text{C-EC} = -15.4 \pm 2.73\text{‰}$) was the main source, while the decline from 2009–2011 points to C3 plants burning ($\delta^{13}\text{C-EC} = -27.5 \pm 2.69\text{‰}$). Notably, the rate of increase in $f_{\text{M-14C}}$ slowed continuously during 2005–2007, and 2012–2014, suggests effective interventions targeting C4 plants emissions during these periods (**Reply Fig. 1b**).

In contrast to EC, global $f_{\text{M-14C}}$ of OC in PM dropped sharply before 2015, then rose, indicating fossil fuel combustion (oil or coal) dominated pre-2015, while biomass burning ($f_{\text{M-14C}} = 1$) became dominant afterward (**Reply Fig. 1c**). The $\delta^{13}\text{C-OC}$ has steadily increased, surpassing the isotopic end-member value for oil, pointing to coal combustion as the main source before 2015, with a shift to C4 plants emissions thereafter (**Reply Fig. 1g**). The slower decline in $f_{\text{M-14C}}$ of OC from 2009 to 2011 suggests effective interventions on coal combustion during this period (**Reply Fig. 1d**).

Global $\delta^{15}\text{N}$ of NH_4^+ in PM decreased from 2001 to 2013, then rose from 2014 to 2020, indicating a shift in main sources of NH_4^+ from isotope-depleted to isotope-enriched ones, e.g., vehicle exhaust (**Reply Fig. 1i**). Notably, during 2017–2020, the rate of $\delta^{15}\text{N-NH}_4^+$ increase slowed continuously, suggesting effective interventions targeting vehicle-derived NH_4^+ -bearing PM (**Reply Fig. 1j**). In contrast, $\delta^{15}\text{N}$ of NO_3^- increased from $-12 \pm 15.9\text{‰}$ to $6.1 \pm 6.5\text{‰}$ between 2001 and 2018, suggesting coal combustion as the main source. Afterward, $\delta^{15}\text{N-NO}_3^-$ declined, pointing to non-coal sources, e.g., gasoline combustion, microbial processes, or biomass burning (**Reply Fig. 1k**). The rate of $\delta^{15}\text{N-NO}_3^-$ increase slowed between 2013 and 2018, indicating effective measures against coal-related NO_3^- in PM (**Reply Fig. 1l**). Similarly, $\delta^{34}\text{S}$ in SO_4^{2-} generally declined, with a brief increase from 2015 to 2016 (**Reply Fig. 1m**). The rate of decline slowed during 2010–2012 and after 2020 (**Reply Fig. 1n**). Based on source $\delta^{34}\text{S}$ values, we suggest that before 2015, coal or biomass combustion was the main source of SO_4^{2-} , effectively controlled between 2010 and 2012. After 2020, isotopic values fell below 2‰, indicating that biomass burning ($\delta^{34}\text{S} = 1.84 \pm 5.41\text{‰}$) became the dominant source and has been progressively managed.

Additionally, there were two turning points in the global trend of $\delta^{87}\text{Sr}$ values in 2005 and 2011. The decline in $\delta^{87}\text{Sr}$ from 2001 to 2005 and from 2011 to the present day indicates that Sr-bearing PM was dominated by isotope-depleted emissions during these periods (**Reply Fig. 1o**). The increase in $\delta^{87}\text{Sr}$ from 2005 to 2011

indicates that Sr-bearing PM mainly originated from isotope-enriched emissions during this period. Additionally, the continuous decrease in the declining rates of $\delta^{87}\text{Sr}$ from 2015 to 2018 indicates the effectiveness of interventions targeting Sr-depleted emissions, e.g., non-exhaust emissions and ore-related emissions (**Reply Fig. 1p**). The $\delta^{202}\text{Hg}$ in PM decreased from 2007 to 2014, indicating that isotope-depleted emissions were the major source of Hg-bearing PM during this period (**Reply Fig. 1q**). From 2010 to 2014, the decreasing rates of $\delta^{202}\text{Hg}$ gradually slowed (**Reply Fig. 1r**), suggesting the effectiveness of interventions targeting Hg-bearing PM from sources such as coal combustion and biomass burning.

For more detailed discussion of other components and the effectiveness of their interventions, please refer to the main text and Supplementary Note 2. Overall, our analysis indicates that effective interventions were only observed for the main sources of PM components, e.g., EC, OC, NO_3^- , NH_4^+ , SO_4^{2-} , Pb, Sr, and Hg. These evaluation results are crucial for developing future PM pollution control strategies.

- 2) Spatially, using the isotopic big data, the stable isotope analysis in R model (SIAR), the mass concentrations of various $\text{PM}_{2.5}$ components, we conducted source tracing analyses for individual $\text{PM}_{2.5}$ compounds. By aggregating the emissions of different components, we estimated the contribution of each subdivided source to overall $\text{PM}_{2.5}$ (see “Methods” for details). Significant variations were observed in the contributions of different sources to $\text{PM}_{2.5}$ and its components (**Reply Fig. 2**). Globally, biomass burning and coal combustion are the main sources, contributing 19.1% and 16.6%, respectively. The main $\text{PM}_{2.5}$ component affected by the two sources is OC, with contributions of 9.7% and 6.0% (**Reply Fig. 2a**). Moreover, vehicle exhaust contributes notably to nitrate in $\text{PM}_{2.5}$, reaching 5.1%. Additionally, some subdivided sources also affect $\text{PM}_{2.5}$ pollution levels by emitting specific components. For instance, liquid fossil fuels and industrial fuel oil combustion contribute 4.0% and 3.3% to $\text{PM}_{2.5}$ via OC and sulfate emission, respectively. Biological processes (e.g., microbial processes) contribute 6.0% to $\text{PM}_{2.5}$ through OC and nitrate emissions. Waste materials, livestock waste, and volatilized fertilizer contribute 1.6% of $\text{PM}_{2.5}$ through the NH_3 emission. These refined source apportionment results are essential for the precise implementation of $\text{PM}_{2.5}$ source control measures.

Regionally, a closer characterization revealed heterogeneity in $\text{PM}_{2.5}$ source composition across Asia, the Americas, and Europe. From 2001 to 2023, coal combustion was the main source of $\text{PM}_{2.5}$ in Asia (**Reply Fig. 2b**), contributing 17.6%, significantly higher than the 12.3% in the Americas and 9.1% in Europe.

This finding aligns with high coal consumption in Asia²². The main components affected by coal combustion in Asia are OC, nitrate, and sulfate, differing from the Americas, where OC and sulfate are the main affected components (**Reply Fig. 2c**), and Europe, where nitrate is the main affected component (**Reply Fig. 2d**). These variations might be attributed to region-specific control processes. Additionally, biomass burning contributes significantly to PM_{2.5} in all the three regions, with the contributions in Europe and the Americas even reaching 22.7% and 29.2%, respectively. OC is the main component affected. Moreover, vehicle exhaust impacts PM_{2.5} components similarly across regions, but the contributions vary: 16.4% in Asia, compared to 8.4% in Europe and 7.7% in the Americas. Noteworthy, some subdivided sources show regional differences in PM_{2.5} components and contributions. For instance, industrial oil combustion contributes substantially more sulfate to PM_{2.5} in Asia than in Europe and the Americas. These differences may be attributed to the industrial structures in different regions.

The related statements have been added in this revised manuscript. Please see page 6 line 124 to page 8 line 186, page 9 line 194 to page 10 line 226, page 30 line 623 to page 31 line 634, page 32 line 635-647, page 40 line 683 to page 43 line 702, page S2 line 22 to page S3 line 65.

Reply Fig. 1. Temporal variations in global PM isotopic fingerprints. a, c, e, g, i, k, m, o, q) Temporal trends of f_{M-14C} for EC (a) and OC (c), $\delta^{13}C$ of EC (e) and OC (g), $\delta^{15}N$ for NH_4^+ (i) and NO_3^- (k), $\delta^{34}S$ of SO_4^{2-} (m), $\delta^{87}Sr$ (o), and $\delta^{202}Hg$ (q) were analyzed using formal bootstrap testing procedures. The abbreviations EC, OC, NH_4^+ , NO_3^- , and SO_4^{2-} represent elemental carbon, organic carbon, ammonium, nitrate, and sulfate respectively. The different color lines in a, c, e, g, i, k, m, o, and q correspond to the liner fits for different periods. b, d, f, h, j, l, n, p, r) Moving subset window analysis of the temporal change trends of $f_{M-14C-EC}$ (b), $f_{M-14C-OC}$ (d), $\delta^{13}C-EC$ (f), $\delta^{13}C-OC$ (h), $\delta^{15}N-NH_4^+$ (j) and $\delta^{15}N-NO_3^-$ (l), $\delta^{34}S$ of SO_4^{2-} (n), $\delta^{87}Sr$ (p), and $\delta^{202}Hg$ (r) over 5-year periods. The analysis step was 1 year. The bars represent isotopic change rates (k) every 5 years. Details on the

amount of isotopic data in each plot are provided in the Source Data file. *** $P < 0.001$, ** $P < 0.01$, * $P < 0.05$. The color scale of the bars correspond to temporal trends, without specific indication.

Reply Fig. 2. Contributions of individual sources to PM_{2.5} and its components in the worldwide (a), Asia (b), the Americas (c), and Europe (d) from 2001–2023. The labels CC, BB, VE, LFF, IOC, NS, BP, WM, NEE, LE, VF, and WI are abbreviations for coal combustion, biomass burning, vehicle exhausts, liquid fossil fuels combustion, industrial oil combustion, natural soil, biological process, waste materials, non-exhaust emissions, livestock emissions, volatilized fertilizer, and waste incinerator, respectively. The uncertainties for these results are provided in the Source Data file. For the specific calculation process, see the “Methods” section. *Note:* in the isotopic source tracing analysis of OC in PM_{2.5}, the liquid fossil fuels combustion is classified as a subdivided source, while OC emissions from vehicle exhaust are not double-counted.

Question: *Second, the architecture of the database has been greatly emphasized as a decentralized blockchain based. However, the dataset itself does not really relate to any scientific findings. It could be a different data paper for journal like ‘Scientific Data’ rather than being emphasized in this paper. The data are presented as Figure 1. It is a bit strange to show the architecture of database rather than the main scientific findings as the first figure.*

Answer: Thanks for your comment. In this revision, we have toned down the statement

on the architecture of the database and highlighted its application in isotopic data management. Moreover, we have moved “Figure 1, IDGAR online portal and blockchain operational principles” to the Extended Data section. Instead, we have included the isotopic fingerprints of different global PM components. This change refocuses our revised manuscript on the spatial and temporal characteristics of PM sources. Please see page 34 line 659 to page 35 line 669.

Specific comments:

Question: *The application of blockchain technology to isotopic data management is interesting, but more details should be provided.*

Answer: Thanks for your comment. We have added detailed statement on the blockchain-based isotopic data management in this revision. Specifically, the curated isotopic data and their details (e.g., sampling date, site location, reference), along with user operations (including authentication information, downloads, and uploads), are transformed into unique 64-character hexadecimal strings using the Secure Hash Algorithm (SHA256, a cryptographic function) and stored as a block in IDGAR (**Reply Fig. 3**)¹⁻³. The SHA256 function is one-way and unbreakable, making the transformed isotopic data tamper-proof. Moreover, each future user action or data update is digitally recorded with timestamps, which are input into the SHA256 function along with the previous block’s string to generate a new block string. Through this iterative hashing pattern, each block is closely linked to all preceding blocks in chronological order, thereby systematically growing the blockchain. Even a minor alteration results in distinct changes in subsequent blocks, enabling precise identification of affected blocks and files by tracing back to the initial block that experienced a hash change. By ensuring data tamper-proofing and increasing trust, IDGAR facilitates the accurate and sustainable application of publicly shared atmospheric isotopic big data for researchers worldwide. Please see page 5 line 91-104, page 34 line 659 to page 35 line 669.

Reply Fig. 3. Workflow of the blockchain-based data management system in the IDGAR. Blockchain technology and the SHA256 function were employed to convert input data into unique digital fingerprints (alphanumeric strings). A consortium chain with 34,993 interconnected blocks, each containing detailed isotopic data and related information, was integrated into the IDGAR. The curated data are stored and managed in a decentralized manner across five node users to deter potential attacks. Users are provided with unique digital identities, generated from their authentication information. Each new operation appends a block that records details such as smart contract information, precise timestamps, isotopic file contents, previous block hashes, and current hashes within an immutable digital fingerprint. The term “DF” represents digital fingerprints.

References:

- 1 Wong, D. R., Bhattacharya, S. & Butte, A. J. Prototype of running clinical trials in an untrustworthy environment using blockchain. *Nat. Commun.* **10**, 917 (2019).
- 2 Chapron, G. The environment needs cryptogovernance. *Nature* **545**, 403-405 (2017).
- 3 Subbiah, V. The next generation of evidence-based medicine. *Nat. Med.* **29**, 49-58 (2023).

Question: *What is the size range of the PM analyzed in this paper? PM₁ or PM_{2.5}?*

Answer: Thanks for your comment. The isotopic signatures of all PM sizes have been compiled in the Isotopic Database for Global Atmospheric Research (IDGAR). Notably, the specific source tracing analysis focuses on PM_{2.5}. In this revision, we have clarified the statement related to the analysis of PM_{2.5}, including the emission constraints of PM_{2.5} using isotopic big data. Please see page 9 line 194, page 11 line 243.

Question: *The description of future simulation using SSP data and GCAM is unclear. It is unclear how the source data provided in this paper can be used for future PM*

simulation.

Answer: Thanks for your comment. In this revision, a set of nested multi-models is used to simulate future PM_{2.5}, and the isotope-based source data provides essential baseline information and correction parameters for their accurate prediction. Specifically, SSP1-1.9 and SSP1-2.6 represent scenarios aimed at controlling future temperature increases to no more than 1.5°C and 2°C, respectively. Coupled with the global change assessment model (GCAM), these scenarios provide insights into the changes in emissions of PM_{2.5} species under different temperature control goals. Additionally, the weather research and forecasting model coupled with chemistry analysis (WRF-Chem) predicts PM_{2.5} pollution levels based on the emission changes. It is worth noting that, due to noticeable uncertainty in emission inventories, the predicted PM_{2.5} pollution levels usually require optimization and correction (**Reply Fig. 4a**). Isotopic-based source tracing results can refine the inventory from a top-down perspective, enhancing the accuracy of PM_{2.5} predictions (**Reply Fig. 4b**). The detailed process is clearly stated in the ‘Methods’ section of the revised manuscript. Please see page 20 line 477-483, page S7 line 97-105.

Reply Fig. 4. Assessment of PM_{2.5} pollution level simulations. Daily mean concentrations of PM_{2.5} in Asia for observation and simulation from April 1, 2024 to April 30, 2024 (**a**), from October 1, 2024 to October 30, 2024 (**b**) were used as a training set to assess the model. **a&b**, the dotted red lines was conducted without incorporating corrections from isotopic tracing results. The dotted blue lines represent the isotopic-guided prediction. The mean bias (MB), root mean square error (RMSE), and index of agreement (IOA) were used to evaluate the predictions against observations. The parameter results indicate that the isotopic-guided predictions are closer to real observations (solid black line).

Response to Reviewer #2:

Question: *In recent years, a variety of data has been conducted from research on aerosols using stable isotope ratios, and this research is generating extremely valuable data. This research is a new challenge using a groundbreaking method that has never*

been used before, using a huge amount of data: 18,760 non-redundant global isotopic observations, including 7,153 sources and 11,607 PM isotopic data published between 1967 and 2024. However, there are several fundamental problems with this paper, and I think that this paper should not be accepted by Nature Communications. The reasons are described below.

Although analysis was done using isotopes of 14 elements (carbon (f_{M-14C} for radioactive; $\delta^{13}C$ for stable), nitrogen ($\delta^{15}N$), oxygen ($\delta^{18}O$), silicon ($\delta^{30}Si$), sulfur ($\delta^{34}S$), iron ($\delta^{56}Fe$), nickel ($\delta^{60}Ni$), copper ($\delta^{65}Cu$), zinc ($\delta^{66}Zn$), strontium ($\delta^{87}Sr$), neodymium ($\delta^{144}Nd$), hafnium ($\delta^{177}Hf$), lead ($^{207}Pb/^{206}Pb$), and mercury ($\delta^{202}Hg$), some elements are in the form of compounds in aerosols. For example, carbon is composed of inorganic carbon and organic carbon, and nitrogen is main composed of different compounds such as nitrate and ammonium salts. In the study of stable isotope aerosols, it was initially impossible to analyze the stable isotope ratios in compounds due to analytical problems, and bulk isotope analysis was the mainstream, but in recent years, isotope analysis of each compound has become the mainstream. In addition, without analyzing the stable isotope ratios of each compound, it is impossible to estimate the origin, and there was a problem in interpreting these data. In this paper, the results are interpreted as the results of the stable isotope ratios of the bulk, but the research is meaningless. If the research were to be conducted based on this idea, it would seem that one approach would be to use only heavy metal isotopes, rather than using isotopes of light elements that form compounds.

Answer: Thanks for your valuable comment. After careful consideration and taking into account your comments, we have made such revisions:

- 1) We have updated isotopic fingerprint data for 34,993 PM compounds and their sources, including 704 f_{M-14C} of organic carbon (OC), 822 f_{M-14C} of element carbon (EC), 467 $\delta^{13}C$ of OC, 921 $\delta^{13}C$ of EC, 4,029 $\delta^{15}N$ of nitrate (NO_3^-), 3,313 $\delta^{15}N$ of ammonium (NH_4^+), and 3,793 $\delta^{34}S$ of sulfate (SO_4^{2-}), which have been updated in the IDGAR database.
- 2) We added a temporal dynamic analysis of PM compound isotopes, including trends and changing rates. Combined with the updated specific source isotopic fingerprints, we reveal the main emission changes and intervention effectiveness for different PM species. Specifically, the global f_{M-14C} of EC in PM increased significantly before 2014, then declined (**Reply Fig. 5a**), indicating that their main emissions shift from non-fossil fuel source (i.e., biomass burning, $f_{M-14C} = 1$) to fossil fuel combustion (coal or oil, $f_{M-14C} = 0$)^{4,5} in 2014. By analyzing the $\delta^{13}C$ -EC trend and source isotopic values, we further identified whether biomass burning before 2014

was dominated by C4 or C3 plants (**Reply Fig. 5e&f**). Specifically, the rise in $\delta^{13}\text{C-EC}$ during 2001–2008 and 2012–2014 suggests C4 plants burning ($\delta^{13}\text{C-EC} = -15.4 \pm 2.73\text{‰}$) was the main source, while the decline from 2009–2011 points to C3 plants burning ($\delta^{13}\text{C-EC} = -27.5 \pm 2.69\text{‰}$). Notably, the rate of increase in $f_{\text{M-14C}}$ slowed continuously during 2005–2007, and 2012–2014 (**Reply Fig. 5b**), suggests effective interventions targeting C4 plants emissions during these periods.

In contrast to EC, global $f_{\text{M-14C}}$ of OC in PM dropped sharply before 2015, then rose (**Reply Fig. 5c**), indicating fossil fuel combustion (oil or coal) dominated pre-2015, while biomass burning ($f_{\text{M-14C}} = 1$) became dominant afterward. The $\delta^{13}\text{C-OC}$ has steadily increased, surpassing the isotopic end-member value for oil (**Reply Fig. 5g&h**), pointing to coal combustion as the main source before 2015, with a shift to C4 plants emissions thereafter. The slower decline in $f_{\text{M-14C}}$ of OC from 2009 to 2011 (**Reply Fig. 5d**) suggests effective interventions on coal combustion during this period.

Moreover, global $\delta^{15}\text{N}$ of NH_4^+ in PM decreased from 2001 to 2013, then rose from 2014 to 2020 (**Reply Fig. 5i**), indicating a shift in main sources of NH_4^+ from isotope-depleted to isotope-enriched ones, e.g., vehicle exhaust. Notably, during 2017–2020, the rate of $\delta^{15}\text{N-NH}_4^+$ increase slowed continuously (**Reply Fig. 5j**), suggesting effective interventions targeting vehicle-derived NH_4^+ -bearing PM. In contrast, $\delta^{15}\text{N}$ of NO_3^- increased from $-12 \pm 15.9\text{‰}$ to $6.1 \pm 6.5\text{‰}$ between 2001 and 2018, suggesting coal combustion as the main source. Afterward, $\delta^{15}\text{N-NO}_3^-$ declined, pointing to non-coal sources, e.g., gasoline combustion, microbial processes, or biomass burning (**Reply Fig. 5k**). The rate of $\delta^{15}\text{N-NO}_3^-$ increase slowed between 2013 and 2018 (**Reply Fig. 5l**), indicating effective measures against coal-related NO_3^- in PM. In addition, a detailed discussion on the main sources of other components (e.g., SO_4^{2-} , metals) and the effectiveness of their interventions is provided in the revised manuscript. Please see page 7 line 146 to page 8 line 186, page S2 line 22 to page S3 line 65.

- 3) As responded above, based on the updated isotopic big data, we performed source tracing analysis for each compound (e.g., OC, EC, metals) in $\text{PM}_{2.5}$. By aggregating the emissions of different components, we estimated the contribution of each subdivided source to overall $\text{PM}_{2.5}$ (**Reply Fig. 2**). Specifically, biomass burning and coal combustion are the main sources, contributing 19.1% and 16.6%, respectively. The main $\text{PM}_{2.5}$ component affected by the two sources is OC, with contributions of 9.7% and 6%. Moreover, vehicle exhaust contributes notably to nitrate in $\text{PM}_{2.5}$, reaching 5.1%. Additionally, liquid fossil fuels and industrial fuel

oil combustion contribute 4% and 3.3% to PM_{2.5} via OC and SO₄²⁻ emission, respectively. Biological processes (including plant debris, fungal bacteria, pollen, particulate matter generated by the oxidation of biogenic volatile organic compound emissions, microbial processes, etc.) contribute 6% to PM_{2.5} through OC and NO₃⁻ emissions. Waste materials, livestock waste, and volatilized fertilizer contribute 2% of PM_{2.5} through the NH₃ emission. Noteworthy, current isotopic-based source tracing analysis mainly relies on components of PM_{2.5} where isotope fingerprints can be obtained. As more component isotopic fingerprints become available, the accuracy and precision of source tracing will further improve. We further analyzed PM_{2.5} source composition across Asia, the Americas, and Europe, and their temporal changes from 2001 to 2023. The detailed calculation process and the related statements are provided in this revised manuscript. Please see page 9 line 194 to page 11 line 242, page 18 line 432 to page 19 line 465, page 30 line 623 to page 31 line 634, page 49 line 719 to page 50 line 724.

Reply Fig. 5. Temporal variations in global PM isotopic fingerprints. a, c, e, g, i, k) Temporal trends of f_{M-14C} for EC (a) and OC (c), $\delta^{13}C$ of EC (e) and OC (g), $\delta^{15}N$ for NH_4^+ (i) and NO_3^- (k), were analyzed using formal bootstrap testing procedures. The abbreviations EC, OC, NH_4^+ , and NO_3^- , represent elemental carbon, organic carbon, ammonium, and nitrate respectively. The

different color lines in **a, c, e, g, i,** and **k** correspond to the liner fits for different periods. **b, d, f, h, j, l**) Moving subset window analysis of the temporal change trends of $f_{M-14C-EC}$ (**b**), $f_{M-14C-OC}$ (**d**), $\delta^{13}C-EC$ (**f**), $\delta^{13}C-OC$ (**h**), $\delta^{15}N-NH_4^+$ (**j**) and $\delta^{15}N-NO_3^-$ (**l**) over 5-year periods. The analysis step was 1 year. The bars represent isotopic change rates (k) every 5 years. Details on the amount of isotopic data in each plot are provided in the Source Data file. *** $P < 0.001$, ** $P < 0.01$, * $P < 0.05$. The color scale of the bars correspond to temporal trends, without specific indication.

Reply Fig. 2. Contributions of individual sources to PM_{2.5} and its components in the worldwide (a), Asia (b), the Americas (c), and Europe (d) from 2001–2023. The labels CC, BB, VE, LFF, IOC, NS, BP, WM, NEE, LE, VF, and WI are abbreviations for coal combustion, biomass burning, vehicle exhausts, liquid fossil fuels combustion, industrial oil combustion, natural soil, biological process, waste materials, non-exhaust emissions, livestock emissions, volatilized fertilizer, and waste incinerator, respectively. The uncertainties for these results are provided in the Source Data file. For the specific calculation process, see the “Methods” section. *Note:* in the isotopic source tracing analysis of OC in PM_{2.5}, the liquid fossil fuels combustion is classified as a subdivided source, while OC emissions from vehicle exhaust are not double-counted.

References:

- Szidat, S. et al. Contributions of fossil fuel, biomass-burning, and biogenic emissions to carbonaceous aerosols in Zurich as traced by 14C. *J. Geophys. Res-Atmos.* 111 (2006).
- Gustafsson, Ö. et al. Brown Clouds over South Asia: Biomass or Fossil Fuel Combustion? *Science* 323, 495-498 (2009).

In addition, the sources of each individual isotope compound are divided into four (biomass burning emissions, coal combustion and industrial emissions, soil-related emissions, and vehicle emissions). However, since there are different sources for each compound, there is a problem with combining these, and the analysis may become meaningless. In addition, the analysis was performed using the isotope mixing model SIAR, but the validity of these is also very uncertain.

Answer: Thank you for your comment. We quite agree that subdividing sources are crucial for accurate PM source tracing. In this revision, we enriched and refined the source information for various PM components, including compounds and metals. Specifically, the isotopic data of vehicle exhausts, coal combustion, biomass burning, C4 plants burning, C3 plants burning, liquid fossil fuels combustion, waste materials, livestock emissions, volatilized fertilizer, microbial processes, diesel vehicle emissions, gasoline vehicle emissions, ore-related emissions, natural soil, industrial oil combustion, non-exhaust emissions, non-exhaust emissions-road paint, non-exhaust emissions-tire, non-exhaust emissions-brake pad and waste incinerator have been considered in this revised manuscript. Moreover, different PM species tend to show different source information. For instance, OC has sources such as coal combustion, C3 plants burning emissions, C4 plants burning emissions, and vehicle exhaust, while EC is not entirely the same, as it also includes emissions from liquefied petroleum gas. Therefore, PM source tracing require multi-species isotopic fingerprints. Notably, most isotopic fingerprints showed notable variations among these sectors for different species ($P < 0.05$), indicating their potential for PM novel source distinction (**Reply Fig. 6**).

Secondly, we are fully aware that reducing the uncertainty of the isotope mixing model SIAR is crucial for isotopic-based source tracing. So, we have taken every opportunity to enhance the validity of using SIAR, e.g.,

- i) According to the characteristics of SIAR (a Bayesian algorithm model coupled with Markov chain Monte Carlo), increasing the data volume and the number of simulations can effectively improve the reliability of the analysis. Thus, we conducted source tracing analysis of PM components using isotopic big data rather than relying on a few scattered data points. Moreover, we set the number of iterations to 1,000,000 with a burn-in of 500,000 in our SIAR simulation. Furthermore, Isospace plots and Markov chain Monte Carlo diagnostic tests (including Gelman-Rubin and Geweke) have been incorporated into the SIAR package to ensure the results are as accurate as possible.
- ii) To further validate of this method, we used f_{M-14C} to evaluate the $\delta^{13}C$ -based

source tracing results of EC in PM_{2.5}. Specifically, we calculated the f_{M-14C} isotopic compositions of EC in PM_{2.5} based on the f_{M-14C} of primary sources and $\delta^{13}C$ -based relative contributions, and compared them with the observed values. As shown in **Reply Fig. 7**, the calculated f_{M-14C} isotopic compositions of EC were consistent with the observed values. The largest net difference between the calculated values and observed data were 0.05, which is smaller than the observed uncertainty of 0.07⁶. This supported the robustness of the $\delta^{13}C$ -based source tracing results.

Based on the isotopic big data of PM and subdivided sources, the stable isotope analysis in R model (SIAR), the mass concentrations of various PM_{2.5} components, we conducted source tracing analyses for individual PM_{2.5} compounds. By aggregating the emissions of different components, we estimated the contribution of each subdivided source to overall PM_{2.5}, and analyzed their spatial-temporal characteristics. The related statements and the detailed verification processes are provided in this revision. Please see page 5 line 108 to page 6 line 116, page 28 line 610 to page 29 line 622, page 18 line 432-438, page 9 line 194 to page 11 line 242, page S4 line 68-75, page S6 line 90-96.

Reply Fig. 6. Isotopic map and statistical results of global PM sources. a-l) Statistical differences in isotopic fingerprints among different emissions. The labels VE, CC, BB, BB_{C4}, BB_{C3}, LFF, WM,

LE, VF, MicP, VE_{diesel}, VE_{gasoline}, OE, NS, IOC, NEE, NEE_{RP}, NEE_T, NEE_{BP}, and WI in (a-l) represent vehicle exhausts, coal combustion, biomass burning, C4 plants burning, C3 plants burning, liquid fossil fuels combustion, waste materials, livestock emissions, volatilized fertilizer, microbial processes, diesel vehicle emissions, gasoline vehicle emissions, ore-related emissions, natural soil, industrial oil combustion, non-exhaust emissions, non-exhaust emissions-road paint, non-exhaust emissions-tire, non-exhaust emissions-brake pad and waste incinerator. The isotopic composition of ammonium (NH₄⁺) and nitrate (NO₃⁻) sources is conventionally expressed in terms of their precursor compounds, namely the isotopic composition of NH₃ and NO_x, respectively^{7,8}. The box extends from the 25% to the 75%, the center line represents the median, and the whiskers indicate the 5% and 95% of the data points. Significance levels are indicated as follows: *****P* < 0.0001, ****P* < 0.001, ***P* < 0.01, **P* < 0.05. *Note:* the isotopic compositions of Zn (i) and Pb (l) from different sources show significant differences (*P* < 0.05), but there are too many statistical results to be clearly labeled in the figure. Therefore, all the data are provided in the Source Data file. Additionally, details on the number of isotopic data points in each window (a-l) are also provided in the Source Data file.

Reply Fig. 7. Verification of EC source apportionment in PM_{2.5} using f_{M-14C} and $\delta^{13}C$ dual isotopic fingerprints. The red bars represent the 6 randomly selected f_{M-14C} observations in EC. The green bars show the calculated f_{M-14C} isotopic compositions of EC in PM_{2.5}, based on the $\delta^{13}C$ of EC sources and their relative contributions. The error bars for both the observed and recalculated results reflect isotopic analysis errors and uncertainties related to source apportionment. For further details, refer to the ‘Methods’ section.

Reference:

- 6 Zhang, Y.-L. et al. Fossil and Nonfossil Sources of Organic and Elemental Carbon Aerosols in the Outflow from Northeast China. *Environ. Sci. Technol.* 50, 6284-6292 (2016).
- 7 Felix, J. D., Elliott, E. M. & Shaw, S. L. Nitrogen Isotopic Composition of Coal-Fired Power Plant NO_x: Influence of Emission Controls and Implications for Global Emission Inventories.

Environ. Sci. Technol. **46**, 3528-3535 (2012).

8 David Felix, J., Elliott, E. M., Gish, T. J., McConnell, L. L. & Shaw, S. L. Characterizing the isotopic composition of atmospheric ammonia emission sources using passive samplers and a combined oxidation-bacterial denitrifier approach. *Rapid Commun. Mass Sp.* **27**, 2239-2246 (2013).

In addition, although the topic of global warming is described as the purpose of the research, it was unclear because there was no description of how it relates to aerosol research.

Answer: Thanks for your comment. To make it clearer, we have added statement on the relationship between PM pollution control and global climate actions targeting the warming control in this revised manuscript. For instance, in addition to current interventions, global ambitious climate actions aimed at warming control will lead to systemic change in PM emissions, e.g., fossil fuels, providing substantial co-benefits for improving air quality⁹⁻¹¹. Moreover, we have toned down the statement on global warming, and focusing on aerosol research. Please see page 11 line 244-245.

Reference:

9 Cheng, J. et al. Pathways of China's PM_{2.5} air quality 2015–2060 in the context of carbon neutrality. *Natl. Sci. Rev.* **8** (2021).

10 Shindell, D. & Smith, C. J. Climate and air-quality benefits of a realistic phase-out of fossil fuels. *Nature* **573**, 408-411 (2019).

11 Scovronick, N. et al. The impact of human health co-benefits on evaluations of global climate policy. *Nat. Commun.* **10**, 2095 (2019).

Reviewer #2 (Remarks on code availability):

I just checked the website.

Answer: Thanks for your comment. The related code and source data have been uploaded as supplementary materials. Please see “code” and “source data” files.

Finally, we thank again the editor and the reviewers for your great efforts on improving the quality of this manuscript.

Thank you very much!

Best wishes,

Yours sincerely,

Dr. Jingwei Zhang
Yunnan University, Kunming, 650500, China
Email: jwzhang@ynu.edu.cn

Dr. Zheng Zong
Shandong University, Qingdao, 266237, China
Email: zzong@sdu.edu.cn

Dr. Dawei Lu
Research Center for Eco-Environmental Sciences
Chinese Academy of Sciences, Beijing 100085, China
Email: dwlu@rcees.ac.cn

Response to Reviewers' Comments

Dear Editor,

We sincerely appreciate the efforts of you and the anonymous reviewers to enhance the quality of our manuscript, titled "Blockchain-based isotopic big data-driven tracing of global PM sources and interventions" (NCOMMS-24-42155A-Z). In response to the reviewers' comments, we have carefully revised the manuscript and Supplement. All changes are highlighted in blue font for clarity. This response letter includes the reviewers' comments in blue italic type, followed by our responses in black font. Each point raised has been thoroughly addressed.

Response to Reviewer #2:

Question:

I highly appreciate the innovative approach taken in this study, particularly the transition from bulk isotopic ratios to compound specific isotopic analysis. This is a significant advancement that enhances our ability to understand the sources of PM_{2.5} and provides a valuable framework for future research. However, I believe that several critical issues remain unresolved. Addressing these concerns will be essential to ensure that the paper meets the standards of Nature Communications.

First, for components such as ammonium and nitrate ions, it is important to consider isotopic fractionation during the conversion of precursor gases (NH₃ and NO_x) into aerosols. These processes can lead to significant isotopic shifts, which must be accounted for when interpreting the sources in a real-world environment. The current manuscript does not appear to discuss this aspect. I suggest focusing on components where isotopic fractionation is negligible (e.g., elemental carbon or heavy metals) or framing the study as a preliminary effort to compile source data rather than attempting to draw definitive conclusions about source apportionment.

Answer: Thanks very much for your valuable comment. We fully agree that isotope fractionation during the formation of key secondary components in PM can result in isotopic shifts, which may impact the accuracy of the apportionment results. We have revised the manuscript in two key aspects. First, we have provided a detailed introduction to the isotopic fractionation correction previously used in our tracing model, an empirical method that is well-documented. Second, following your suggestion, we have toned down the quantitative discussion of tracing results for components prone to isotopic fractionation (e.g., ammonium, nitrate ions) in the revised manuscript. The specific changes are detailed as follows:

(i) For NH₄⁺, isotopic fractionation was quantified using a widely accepted method for

particulate NH_4^+ (Pan et al., 2016; Zhang et al., 2020):

$$35 \quad \delta^{15}\text{N-NH}_3 = \delta^{15}\text{N-NH}_4^+ - \varepsilon_{\text{NH}_4^+/\text{NH}_3} \times (1 - f) \quad (1)$$

where, $\delta^{15}\text{N-NH}_3$ and $\delta^{15}\text{N-NH}_4^+$ represent the isotopic values of gaseous NH_3 and its converted
particulate NH_4^+ , respectively. $\varepsilon_{\text{NH}_4^+/\text{NH}_3}$ denotes the nitrogen equilibrium isotope fractionation
factor, calculated as $(12.4678 \times 1000/T) - 7.6694$, where T represents the ambient temperature
in Kelvin. The parameter f represents the molar conversion ratio, defined as $[\text{NH}_4^+ / (\text{NH}_4^+ +$
$\text{NH}_3)]$. Chen et al. (2022) provided a novel insight by demonstrating that f values exhibit
regional and temporal variations on a global scale (**Reply Fig. 1**). In our study, we selected
appropriate f values tailored to each sampling location, considering both geographic and
temporal contexts.

For NO_3^- , the isotope fractionation of NO_3^- ($\Delta^{15}\text{N}$) during its formation was quantified by
accounting for the combined effects of two primary pathways involving O_3 and $\bullet\text{OH}$ (Zong et
al., 2017):

$$47 \quad \Delta^{15}\text{N} = \gamma \times \Delta(\delta^{15}\text{N-NO}_3^-)_{\text{OH}} + (1 - \gamma) \times (\delta^{15}\text{N-NO}_3^-)_{\text{O}_3} \quad (2)$$

where, γ represents the contribution ratio of the $\bullet\text{OH}$ pathway. Isotopic fractionation occurs
during the reaction between NO_2 and photochemically produced $\bullet\text{OH}$, denoted as $\Delta(\delta^{15}\text{N-}$
$\text{NO}_3^-)_{\text{OH}}$. The remaining fractionation is attributed to the hydrolysis of N_2O_5 , expressed as
$\Delta(\delta^{15}\text{N-NO}_3^-)_{\text{O}_3}$. Given that the $\text{NO}_3^- + \text{hydrocarbon (HC)}$ pathway is a minor contributor on a
global scale (Alexander et al., 2009), its influence on fractionation was ignored. Moreover, our
previous research demonstrated a significant linear relationship between γ and latitude (Zong
et al., 2020). In this study, γ was estimated based on the latitude of each sampling location. In
addition, the $\Delta(\delta^{15}\text{N-NO}_3^-)_{\text{OH}}$ can be calculated using a mass-balance according to Zong et al.
(2017):

$$57 \quad \Delta(\delta^{15}\text{N-NO}_3^-)_{\text{OH}} = 1000 \times \left[\frac{(^{15}\alpha_{\text{NO}_2/\text{NO}} - 1)(1 - f_{\text{NO}_2})}{(1 - f_{\text{NO}_2}) + (^{15}\alpha_{\text{NO}_2/\text{NO}} \times f_{\text{NO}_2})} \right] \quad (3)$$

where $^{15}\alpha_{\text{NO}_2/\text{NO}}$ is the equilibrium isotope fractionation factor between NO_2 and NO , which is
a temperature-dependent function (**refer to equation 5 and Table 1**). The parameter f_{NO_2}
denotes the fraction of NO_2 in total NO_x with reported values ranging from 0.2 to 0.95 (Zong
et al., 2017). Similarly, $\Delta(\delta^{15}\text{N-NO}_3^-)_{\text{O}_3}$ can be determined from the following equation:

$$62 \quad \Delta(\delta^{15}\text{N-NO}_3^-)_{\text{O}_3} = 1000 \times (15\alpha_{\text{N}_2\text{O}_5/\text{NO}_2} - 1) \quad (4)$$

where $^{15}\alpha_{\text{N}_2\text{O}_5/\text{NO}_2}$ refers to the equilibrium isotope fractionation factor between N_2O_5 and NO_2 ,
which is also a temperature-dependent function (**see equation 5 and Reply Table 1**). For the
$^{15}\alpha_{\text{NO}_2/\text{NO}}$ and $^{15}\alpha_{\text{N}_2\text{O}_5/\text{NO}_2}$ in these equations, the $^{m}\alpha_{X/Y}$ is a function of temperature, and can be
expressed as:

$$67 \quad 1000(^m\alpha_{X/Y} - 1) = \frac{A}{T^4} \times 10^{10} + \frac{B}{T^3} \times 10^8 + \frac{C}{T^2} \times 10^6 + \frac{D}{T} \times 10^4 \quad (5)$$

where A, B, C, and D are experimental constants (**Reply Table 1**) over the temperature range
 of 150-450 K. For a comprehensive and detailed discussion of these isotopic fractionations,
 you can find the information in the papers by Zong et al. (2017) and Walters et al. (2015, 2016).

To minimize model uncertainty arising from variations in the f_{NO_2} value, an iterative
 approach was applied, using a simulation step of 0.01 times $\Delta^{15}\text{N}$. The results indicated that
 when the model used 0.66 times $\Delta^{15}\text{N}$, the probability distribution of source contributions
 exhibited the lowest variance. This value was therefore identified as the most likely solution.

For SO_4^{2-} , previous research has identified a strong linear relationship between the isotope
 fractionation factor and ambient temperature during SO_4^{2-} formation (Lin et al., 2021; **Reply**
 **Fig. 2**). In this study, we utilized this relationship to quantify isotopic fractionation
 characteristics associated with SO_4^{2-} production. By combing recorded ambient temperatures
 for each sampling campaign with a Rayleigh distillation model, we calculated the isotopic
 composition of precursor to particulate SO_4^{2-} :

$$81 \quad \delta^{34}\text{SO}_2 = \delta^{34}\text{SO}_4^{2-} \times \left(\frac{1-f}{f^{34}\alpha_{\text{Sg}\rightarrow\text{p}} - 1} \right) \quad (6)$$

where $\delta^{34}\text{S-SO}_2$ and $\delta^{34}\text{S-SO}_4^{2-}$ represent the isotopic values of gaseous SO_2 and its resulting
 particulate SO_4^{2-} , respectively. The term $\alpha^{34}\text{S}_{\text{g}\rightarrow\text{p}}$ refers to the isotope fractionation factor
 between gaseous SO_2 and particulate SO_4^{2-} , while f denotes the fraction of SO_2 remaining in
 gas phase, with reported values ranging from 0.1 to 0.9 (Fan et al., 2020). To further reduce
 uncertainties associated with f , we employed an iterative modelling approach. Our results
 revealed that when the model used 0.51 times isotope fractionation, the probability distribution
 of source contributions was more reasonable.

For OC, whose $\delta^{13}\text{C}$ isotopic fractionation effect is corrected using $f_{\text{M-14C}}$ (fractions of
 modern carbon, the isotopic characteristics of ^{14}C) (Zhao et al., 2022; Vlachou et al., 2018).
 The theoretical value of $\delta^{13}\text{C}$ in OC is primarily calculated through the mass balance of ^{14}C ,
 and the correction factor is obtained by comparing this with the observed $\delta^{13}\text{C}$ value in OC.
 The specific calculation formula is as follows:

$$94 \quad f_{\text{nf}} \times \delta^{13}\text{C}_{\text{nf}} + f_{\text{coal}} \times \delta^{13}\text{C}_{\text{coal}} + f_{\text{liq.fossil}} \times \delta^{13}\text{C}_{\text{liq.fossil}} = \delta^{13}\text{C}_{\text{sample}} + \beta \quad (7)$$

where $\delta^{13}\text{C}_{\text{nf}}$, $\delta^{13}\text{C}_{\text{coal}}$, $\delta^{13}\text{C}_{\text{liq.fossil}}$, and $\delta^{13}\text{C}_{\text{sample}}$ represent the $\delta^{13}\text{C}$ values for non-fossil sources,
 coal, liquid fossil fuels, and the OC sample, respectively. The f_{nf} , f_{coal} , and $f_{\text{liq.fossil}}$ denote the
 contributions of non-fossil fuels, coal, and liquid fossil fuels, respectively. The specific
 calculation process for these factors can be referenced in these studies (Hou et al., 2021; Levin
 et al., 2010; Ni et al., 2020). The obtained β under different conditions will be incorporated into
 the tracing model to implement the correction of OC isotopic fractionation effects.

**(ii)** To maintain rigor, we have concentrated the discussion on components with negligible
 isotopic fractionation in this revision. For instance, globally, biomass burning and vehicle

exhaust are the leading contributors to EC in PM_{2.5}, accounting for 2.17 ± 0.09 % and $2.23 \pm$
0.09 % of the total mass of PM_{2.5}, respectively, through EC emissions. Natural soil and coal
combustion are the major sources of Si in PM_{2.5}, with Si emissions from these sources
contributing 1.99 ± 0.08 % and 1.22 ± 0.06 %, respectively, to the overall PM_{2.5} mass. Non-
exhaust emissions are a key source of metal elements in PM_{2.5}, especially Fe and Zn, with Fe
and Zn emissions contributing 0.26 ± 0.02 % and 0.15 ± 0.01 % to the total mass of PM_{2.5},
respectively. Moreover, regionally, biomass burning in Asia contributes approximately $1.29 \pm$
0.82 % of PM_{2.5} through EC emission, which is lower than its contribution in the Americas
(6.12 ± 1.70 %) and Europe (5.06 ± 4.30 %).

Moreover, we highlight the importance of isotopic correction in this revision. For instance,
noteworthy, although the isotopic fractionation correction of certain components, such as NH₄⁺
and NO₃⁻, during the conversion process has been accounted for in the MixSIAR model (see
“Methods” and Supplementary Note 3 for details), future in-depth studies on the isotopic
fractionation mechanisms of specific transformation reactions under real-world environmental
conditions will further improve the accuracy of source apportionment results.

The related statements have been added in this revised manuscript. Please see page 9 line
200-205, page 19 line 446-448, page S3 line 66 to page S6 line 133, page 9 line 212 to page 10
line 216, page 10 line 219-221, page 10 line 227-233.

**Reply Table 1. The test constant of A, B, C, D over the settled temperature range (150-450K).**

^m α _{X/Y}	A	B	C	D	equation
¹⁵ NO ₂ /NO	3.8834	-7.7299	6.0101	-0.17928	3
¹⁵ N ₂ O ₅ /NO ₂	0.69398	-1.9859	2.3876	0.16308	4

Reply Fig. 1. Time series of global $f_{p\text{-NH}_4^+}$ revealed by Chen et al. (2022).

Reply Fig. 2. Scatter plot of $\alpha^{34}\text{S}_{g\rightarrow p}$ against the ambient temperature, as reported by Lin et al. (2021).

Reference:

- Pan, Y. P. *et al.* Fossil fuel combustion-related emissions dominate atmospheric ammonia sources during severe haze episodes: Evidence from ^{15}N -stable isotope in size-resolved aerosol ammonium. *Environ. Sci. Technol.* **50**, 8049-8056 (2016).
- Zhang, Y. Y. *et al.* Persistent nonagricultural and periodic agricultural emissions dominate sources of ammonia in urban Beijing: Evidence from ^{15}N stable isotope in vertical profiles. *Environ. Sci. Technol.* **54**, 102-109 (2020).
- Chen, Z. L. *et al.* Significant contributions of combustion-related sources to ammonia emissions. *Nat. Commun.* **13**, 7710 (2022).
- Zong, Z. *et al.* First assessment of NO_x sources at a regional background site in north China using isotopic analysis linked with modeling. *Environ. Sci. Technol.* **51**, 5923-5931 (2017).
- Alexander, B. *et al.* Quantifying atmospheric nitrate formation pathways based on a global model of the oxygen isotopic composition ($\Delta^{17}\text{O}$) of atmospheric nitrate. *Atmos. Chem. Phys.* **9**, 5043-5056 (2009).

- Zong, Z. *et al.* Dual-modelling-based source apportionment of NO_x in five Chinese megacities: Providing
the isotopic footprint from 2013 to 2014. *Environment International* **137**, 105592 (2020).
- Walters, W. W. & Michalski, G. Theoretical calculation of nitrogen isotope equilibrium exchange
fractionation factors for various NO_y molecules. *Geochim. Cosmochim. Ac.* **164**, 284-297 (2015).
- Walters, W. W. & Michalski, G. Theoretical calculation of oxygen equilibrium isotope fractionation factors
involving various NO_y molecules, OH, and H₂O and its implications for isotope variations in
atmospheric nitrate. *Geochim. Cosmochim. Ac.* **191**, 89-101 (2016).
- Lin, Y. -C., Yu, M. Y., Xie, F., & Zhang, Y. L. Anthropogenic emission sources of sulfate aerosols in
Hangzhou, East China: insights from isotope techniques with consideration of fractionation effects
between gas-to-particle transformations. *Environ. Sci. Technol.* **56**, 3905-3914 (2022).
- Fan, M.-Y. *et al.* Roles of sulfur oxidation pathways in the variability in stable sulfur isotopic composition
of sulfate aerosols at an urban site in Beijing, China. *Environ. Sci. Technol. Lett.* **7**, 883-888 (2020).
- Zhao, H. Y. Z. *et al.* Measurement report: source apportionment of carbonaceous aerosol using dual-carbon
isotopes (¹³C and ¹⁴C) and levoglucosan in three northern Chinese cities during 2018–2019. *Atmos.*
*Chem. Phys.* **22**, 6255-6274 (2022).
- Vlachou, A. *et al.* Advanced source apportionment of carbonaceous aerosols by coupling offline AMS and
radiocarbon size-segregated measurements over a nearly 2-year period. *Atmos. Chem. Phys.* **18**, 6187-
6206 (2018).
- Hou, S. Q. *et al.* Source apportionment of carbonaceous aerosols in Beijing with radiocarbon and organic
tracers: insight into the differences between urban and rural sites. *Atmos. Chem. Phys.* **21**, 8273-8292
(2021).
- Levin, I. *et al.* Observations and modelling of the global distribution and long-term trend of atmospheric
¹⁴CO₂. *Tellus B.* **62**, 26-46 (2010).
- Ni, H. *et al.* Measurement report: dual-carbon isotopic characterization of carbonaceous aerosol reveals
different primary and secondary sources in Beijing and Xi'an during severe haze events. *Atmos. Chem.*
*Phys.* **20**, 16041-16053 (2020).

*Second, I found it unclear how source contributions were calculated in cases where source-*
*specific data are missing. For example, global δ¹⁵N values for ammonia from biomass burning*
*are not yet well-established. Without these critical data, the reliability of the source*
*apportionment results is questionable. This point requires clarification or additional discussion*
*to justify the methodology.*

**Answer:** Thanks very much for your valuable comment. At present, the δ¹⁵N values for
ammonia from biomass burning is available only for Asia (Li et al., 2023), with corresponding
data for the Americas and Europe absent. To estimate global PM ammonium emissions,
previous studies have attempted to use nitrogen isotope endmember values from Asian biomass
as a proxy for global endmember data in their analysis (Chen et al., 2022). As such, we used
Asian values as global proxies, without taking into account the potential subtle differences in
ammonium isotopes from biomass burning across regions for the time being. Meanwhile, in
this revision, we clarified this point and highlight that as more component isotopic fingerprints
become available, e.g., NH₄⁺ in the Americas and Europe, the accuracy and precision of source
tracing will further improve. To enhance rigor, we intentionally refrained from engaging in a
detailed discussion of the regional source characteristics of NH₄⁺ and chose not to quantify its
contribution to specific sources in the Americas and Europe (**Reply Fig. 3c-d**). Following your
earlier suggestion, we have focused the quantitative source apportionment discussion on PM
components shared by all three regions. For instance, the same type of sources, such as biomass

burning, show varying contributions to PM_{2.5} components across Asia, the Americas, and
 Europe. Specifically, biomass burning in Asia contributes approximately 1.29 ± 0.82 % of
 PM_{2.5} through EC emissions, which is lower than its contribution in the Americas ($6.12 \pm$
 1.70 %) and Europe (5.06 ± 4.30 %). Noteworthy, even when excluding ammonium, biomass
 burning continues to be the largest source of PM_{2.5} emissions in both the Americas and Europe.
 Moreover, we directly used the ammonium inventory instead of the isotope-based ammonium
 inventory to predict future PM_{2.5} pollution trends. The variation is small, almost negligible
 (**Reply Fig. 4**). Therefore, this modification does not affect the overall discussion in the paper.
 The related statements have been added in this revised manuscript. Please see page 32 line 664-
 666, page 11 line 251-253, page 9 line 212 to page 10 line 216, page 10 line 219-221, page 10
 line 227-233.

 **Reply Fig. 3 Contributions of individual sources to PM_{2.5} and its components in the worldwide (a),**
 **Asia (b), the Americas (c), and Europe (d) from 2001–2023.** The labels CC, BB, VE, LFF, IOC, NS, BP,
 WM, NEE, LE, VF, and WI are abbreviations for coal combustion, biomass burning, vehicle exhausts, liquid
 fossil fuels combustion, industrial oil combustion, natural soil, biological process, waste materials, non-
 exhaust emissions, livestock emissions, volatilized fertilizer, and waste incinerator, respectively. Biological
 processes include plant debris, fungal bacteria, pollen, particulate matter generated by the oxidation of
 biogenic volatile organic compound emissions, microbial processes. The uncertainties for these results are
 provided in the Source Data file. For the specific calculation process, see “Methods” section. *Note:* in the
 isotopic source tracing analysis of OC in PM_{2.5}, the liquid fossil fuels combustion is classified as a subdivided
 source, while OC emissions from vehicle exhaust are not double-counted. Due to the absence of the $\delta^{15}\text{N}$
 values for ammonia from biomass burning in the Americas and Europe, the global source apportionment
 study relied on the $\delta^{15}\text{N}$ values for ammonia from Asian biomass burning to estimate global values.

 **Reply Fig. 4 Projected PM_{2.5} pollution trends until 2100 for the Americas.** The scenarios, SSP1-1.9 and
 SSP1-2.6, combine Shared Socioeconomic Pathways (SSP) with Representative Concentration Pathways
 (RCP), which correspond to future radiative forcing and their climate impacts. Specifically, SSP1-1.9 and
 SSP1-2.6 aim to limit global warming to 1.5°C and 2°C, respectively. Error bars represent mean ± s.d. ($n =$
 3). For more details about the scenario settings in the PM pollution simulations, see the “Methods” section.
 These projected results by 2100 are close to the WHO guideline limit of 5 µg/m³, but have not yet been
 achieved

**References:**

Li, Y. Z. *et al.* Apportioning atmospheric ammonia sources across spatial and seasonal scales by their isotopic
 fingerprint. *Environ. Sci. Technol.* 2023, **57**, 16424-16434 (2023).
 Chen, Z. L. *et al.* Significant contributions of combustion-related sources to ammonia emissions. *Nat.*
 *Commun.* **13**, 7710 (2022).

*Third, the study relies exclusively on the SIAR model for source apportionment. While this is a*
 *well-established approach, there are other models available, such as MixSIR or FRUITS, which*
 *may offer complementary insights. Have the authors considered comparing these models, or at*
 *least discussing their relative strengths and limitations? Additionally, the paper does not*
 *provide a clear discussion of the uncertainties in the calculated source contributions. Including*
 *an analysis of uncertainties would significantly strengthen the robustness of the results.*

*In conclusion, this paper makes an important contribution to the field by introducing a*
 *framework for component-level isotopic analysis. However, resolving the above issues will be*
 *necessary to enhance the scientific rigor and clarity of the study. Addressing these points would*
 *also help the paper better align with the expectations of a journal like Nature Communications.*

**Answer:** Thanks very much for your valuable comment. After careful consideration and taking
 into account your comments, we have made such revisions:

First, we need to clarify the previously ambiguous statements. The tracing model we used
 is MixSIAR, a Bayesian tracing model, rather than SIAR. For further details, please refer to
 the source code included in this submission. Moreover, we have made revisions throughout the

manuscript.

Second, in this revision, we compared the performance of MixSIAR, MixSIR, and
FRUITS in the tracing study. We selected six PM_{2.5} pollution events that reported the isotopic
compositions of EC for both ¹⁴C (f_{M-14C}) and ¹³C ($\delta^{13}C$). Using these data, we calculated the
relative contributions of individual sources to EC based on $\delta^{13}C$ values with MixSIAR, MixSIR,
and FRUITS. Then, based on the $\delta^{13}C$ -based contributions, we calculated the f_{M-14C} of EC in
PM_{2.5} and compared these with the observed f_{M-14C} values. As shown in **Reply Fig. 5**, the f_{M-14C}
values calculated with MixSIAR show a smaller difference from the observed values
compared to those from MixSIR and FRUITS, supporting the robustness of the MixSIAR
model. All three models are Bayesian-based and suitable for isotopic tracing analysis. Among
these, MixSIAR combines the MixSIR and SIAR models. Its advantage is that it incorporates
both fixed and random effects as covariates, enabling the explanation of variability in mixing
proportions. MixSIR is implemented using the MATLAB language. Its advantage lies in its
ability to account for multi-source isotopic characteristics and isotopic discrimination, though
it falls short in fully addressing data uncertainty. FRUITS is primarily utilized for analyzing
food source contributions. Its advantage lies in its ability to correct isotopic pathways for
proteins, fats, and other substances. However, when applied to PM_{2.5} tracing, this may
necessitate further optimization.

Third, the uncertainties in the calculated source contributions have been added in this
revision. The uncertainty in source apportionment was derived from the MixSIAR model
analysis and the error propagation formula. The detailed calculation process is as follows:

$$266 \quad \sigma_{Ssum} = \sqrt{\sum_{i=1}^n \left[(X_E \times C_{ES}) \times \sqrt{\left(\frac{\sigma_{X_E}}{X_E}\right)^2 + \left(\frac{\sigma_{C_{ES}}}{C_{ES}}\right)^2} \right]}$$

where σ_{Ssum} represents the standard deviation of the total contributions from source ‘S’ to PM_{2.5}.
X_E denotes the mass concentration of component ‘E’ in PM_{2.5}, and C_{ES} represents the relative
contribution of source ‘S’ to component ‘E’ in PM_{2.5}, as calculated by MixSIAR. Additionally,
σ_{X_E} indicates the uncertainty in the mass concentration of component ‘E’ in PM_{2.5}, while $\sigma_{C_{ES}}$
reflects the uncertainty in the C_{ES} values derived from MixSIAR.

The related statements have been added in this revised manuscript. Please see page 9 line 195-
196, page 18 line 436-437; page 9 line 205-208, page 20 line 479-481, page S6 line 134-150,
page S9 line 175-178; page 9 line 198-200, page 19 line 464 to page 20 line 471, page 9 line
212-215, page 10 line 219-221, page 10 line 230-233.

Reply Fig. 5. Verification of EC source apportionment in PM_{2.5} using f_{M-14C} and $\delta^{13}C$ dual isotopic fingerprints. The error bars for both the observed and recalculated results.

Finally, we thank again the editor and the reviewers for your great efforts on improving the quality of this manuscript.

Thank you very much!

Best wishes,

Yours sincerely,

Dr. Jingwei Zhang

Yunnan University, Kunming, 650500, China

Email: jwzhang@ynu.edu.cn

Dr. Zheng Zong

Shandong University, Qingdao, 266237, China

Email: zzong@sdu.edu.cn

Dr. Dawei Lu

Research Center for Eco-Environmental Sciences

Chinese Academy of Sciences, Beijing 100085, China

Email: dwlu@rcees.ac.cn

Response to Reviewers' Comments

Response to Reviewer #2:

Question:

I highly appreciate the authors' efforts in considering isotopic fractionation during secondary formation and their attempts to explore multiple isotopic mixing models. These aspects significantly enhance the scientific rigor of the study and should be recognized. However, several concerns remain unresolved. Unless these issues are properly addressed, I believe acceptance in Nature Communications will be challenging.

Major Consideration: Reliability of Source Data

I have significant concerns regarding the reliability of the source data. For example, the nitrogen stable isotope ratios ($\delta^{15}\text{N}$) of ammonia gas varies significantly depending on the sampling method. It has been demonstrated that passive samplers and active samplers can yield $\delta^{15}\text{N}$ differences as large as 15‰, as shown by Pan et al. (2020) and Kawashima et al. (2021). Such methodological discrepancies must be carefully considered when integrating source data from multiple studies. Moreover, similar issues may exist for other source data beyond ammonia. This discrepancy is critical because it directly affects the reliability of source apportionment models. However, the manuscript does not indicate whether such methodological differences were considered when integrating source data from multiple studies. Furthermore, similar concerns apply not only to ammonia but also to other source data included in the database. If the same level of scrutiny is not applied across all source categories, the resulting database may lack scientific rigor and lead to misleading conclusions. A comprehensive assessment of data reliability is essential to ensure that the isotopic source apportionment framework remains valid.

To improve the robustness of the study, I strongly recommend that the authors explicitly clarify in manuscript or database:

1. What sampling and analytical methods were used for all source data? – Were the reported isotope values derived from a consistent approach, or do they include data obtained under different methodologies without correction?

2. Which specific sources were included in the dataset? – A table summarizing the locations, time periods, and measurement techniques for each source type (not just ammonia but also elemental carbon, sulfate, metals, etc.) would improve transparency.

3. How methodological bias was addressed? – If different studies employed different sampling or analytical techniques, were correction factors applied? If not, how do the authors justify

*integrating isotopic data that may not be directly comparable?"*

**Answer:** Thanks very much for your valuable comment. After careful consideration and taking
into account your comments, we have revised the manuscript accordingly:

i) The detailed sampling and analytical methods used for all source data are well summarized
in the **Supplementary Table 4** and **Supplementary Table 5** in this revision. Briefly, the
sampling of NH₃ from sources is categorized into active and passive sampling. The results of
passive sampling were corrected according to the correction method described below^[1]. Source
emissions of NO_x are collected through active sampling using the standard solution absorption
method^[2]. Similarly, almost source emissions of SO₂ are also collected via active sampling
with solution absorption. Only one study used passive sampling for SO₂^[3], and we excluded its
reported values from S source tracing in the revised manuscript. Aside from NH₃, NO_x, and
SO₂, other PM components (such as OC, EC, Si, and metals) included in the database are not
sampled individually but are collected through active sampling of source emissions, followed
by the analysis of the specific isotopic fingerprint characteristics of the different components
within the particulate matter. Additionally, different components may be analyzed using various
methods. For instance, nitrogen isotope measurement of NH₃ is conducted using both chemical
conversion and bacterial conversion methods; for Sr isotopic measurement might be performed
using thermal-ionization mass spectrometry (TIMS) or multi-collector inductively coupled
plasma mass spectrometry (MC-ICP-MS), both of which undergo strict quality control.

ii) The dataset includes isotopic signatures from various sources, e.g., biomass burning, coal
combustion, vehicle exhausts, waste materials, volatilized fertilizer, livestock emissions,
microbial processes, industrial oil combustion, natural soil. A summary table detailing the
locations, time periods, and measurement techniques for each source type has been included as
**Supplementary Table 4** and **Supplementary Table 5** for improved transparency.

iii) To address potential methodological biases, we standardized the results obtained from
different sampling and analytical methods that required correction. Specifically, for ammonia
source apportionment applications, data correction for passive sampling is performed by
adding 15.4%, following commonly used methods in the literature^[1]. For NO_x and SO₂, the
recovery of active sampling is close to 100%, and hence no correction was needed. Since only
one study used passive sampling for SO₂, we excluded its reported values from S source tracing
in the revised manuscript to ensure data standardization and the reliability of the research
findings. For other PM components (such as OC, EC, Si, and metals), as they are primary
particulate matter collected directly from the source and analyzed for their isotopic composition,
they can directly reflect the emissions from the source. Regarding to different analytical
methods, the isotopic results from different analytical methods are undergo strict quality
control and reported with reference to international standard reference materials, ensuring

consistency and standardization. Regarding different standard reference materials, such as Zn,
the Zn isotopic composition is typically expressed in per mil (‰) relative to the “zero point”
of the isotopic standard reference material, IRMM-3702. Notably, some publications report the
Zn isotopic composition relative to JMC 3-0749L Lyon (another reference material). The Zn
isotopic composition of JMC 3-0749L Lyon is +0.32‰ relative to IRMM-3702. For clarity, the
Zn isotopic signatures presented here have been converted to be relative to IRMM-3702.

Moreover, all results and figures related to this correction have been updated in this
revision. We have also clarified these points in the revised manuscript and highlight that, with
the emergence of more precise cross-validation methods, the data in this database needs to be
standardized and updated. The related statements have been added in this revision, please see
page 15 line 349 to page 16 line 379, page 29 line 658 to page 30 line 670, page 33 line 683-
697, page 34 line 699-708, page S23 line 220 to page S24 line 225, page S64 line 293 to page
S288. The **Supplementary Table 4** is attached to this response letter for your convenience.
However, due to its large file size, **Supplementary Table 5** could not be uploaded as an
attachment to this response letter. Please refer to the Supplementary Information materials
instead.

**References:**

- [1] Gu, M. et al. Vehicular Emissions Enhanced Ammonia Concentrations in Winter Mornings: Insights from
Diurnal Nitrogen Isotopic Signatures. *Environ. Sci. Technol.* 56, 1578-1585 (2022).
[2] Shi, Y. et al. Stable nitrogen isotope composition of NO_x of biomass burning in China. *Sci. Total Environ.*
803, 149857 (2022).
[3] Hong, Y., Zhang, H. & Zhu, Y. Sulfur isotopic characteristics of coal in China and sulfur isotopic
fractionation during coal-burning process. *Chin. J. Geochem.* 12, 51-59 (1993).

*Further Consideration: The Need for a Correctable Database*

*Given the well-documented impact of sampling and analytical techniques on isotope ratios*
*(Pan et al., 2020; Kawashima et al., 2021), I also suggest that the authors discuss the potential*
*for updating and refining the database as new, more accurate source data become available. A*
*rigid, static database that does not allow for corrections in light of new findings may introduce*
*long-term issues in source apportionment. The authors should consider implementing a*
*mechanism for ongoing validation and revision of the database to ensure its long-term*
*scientific reliability.*

*Pan, Y., Gu, M., Song, L., Tian, S., Wu, D., Walters, W. W., . . . Wang, Y. (2020). Systematic low*
*bias of passive samplers in characterizing nitrogen isotopic composition of atmospheric*
*ammonia. Atmospheric Research, 243,*
*105018. <https://doi.org/10.1016/j.atmosres.2020.105018>.*

*Kawashima, H., Ogata, R., & Gunji, T. (2021). Laboratory-based validation of a passive*
*sampler for determination of the nitrogen stable isotope ratio of ammonia gas. Atmospheric*

*Environment*, 245, 118009. <https://doi.org/10.1016/j.atmosenv.2020.118009>.

**Answer:** Thanks very much for your valuable comment. We agree that the accuracy of isotopic
data is crucial for source apportionment, and acknowledge the need for continual updates to
the database as new and more accurate source data become available in this revision. The
corrected data in this revision has been updated in our database. Moreover, we have updated
the source isotope data upload template to require detailed records of the sampling locations,
time periods, and measurement techniques for each source type. Additionally, we have added
the statement “Please upload the source component isotope data from active sampling,
including detailed records of the sampling locations, time periods, and measurement techniques
for each source type. If the data were obtained through passive sampling, please contact
dwlu@rcees.ac.cn to obtain the calibration conversion method.” Finally, we plan to verify the
entire database every two years to ensure its continued scientific reliability over time.

Finally, we thank again the editor and the reviewers for your great efforts on improving the
quality of this manuscript.

Thank you very much!

Best wishes,

Yours sincerely,

Dr. Jingwei Zhang

Yunnan University, Kunming, 650500, China

Email: jwzhang@ynu.edu.cn

Dr. Zheng Zong

Shandong University, Qingdao, 266237, China

Email: zzong@sdu.edu.cn

Dr. Dawei Lu

Research Center for Eco-Environmental Sciences

Chinese Academy of Sciences, Beijing 100085, China

Email: dwlu@rcees.ac.cn

Supplementary Table 4. Summary of Source Emission Sampling and Isotopic Analysis.

Isotope	Content	Source type	Sampling method	Analytical method	Quality Control Materials
$\delta^{15}\text{N}$	NH_3	Biomass Burning	1. Passive sampling (Corrected by adding 15.4%) 2. Active sampling	Chemical method ^[821,822] ;	IAEA-N, USGS25, USGS26 ^[a]
		Coal Combustion		Biological method ^[823] ;	
		Vehicle Exhausts		analyzed by Isotope Ratio Mass	
		Waste Materials		Spectrometer (including IRMS, PT-	
		Volatilized Fertilizer		IRMS, and CF-IRMS)	
Livestock Emissions					
$\delta^{15}\text{N}$	NO_x	Biomass Burning	Active sampling	Biological method ^[823] , analyzed by	IAEA-N3, USGS32, USGS34, USGS35 ^[b]
		Coal Combustion		Isotope Ratio Mass Spectrometer	
		Microbial Processes		(including IRMS, EA-IRMS, and CF-	
		Vehicle Exhausts		IRMS)	
$\delta^{34}\text{S}$	SO_2	Biomass Burning	1. Active sampling 2. Passive sampling (only 1 publication, the data was not used for source apportionment)	Isotope Ratio Mass Spectrometer or VG Iso-gas Mass Spectrometer	IAEA S-1, IAEA S-2, IAEA- SO5, IAEA-SO6, LTB-2, LTB-5, CSIRD, NBS127 ^[c]
		Coal Combustion			
		Industrial Oil Combustion			
		Natural Soil			
		Vehicle Exhausts			

$\delta^{13}\text{C}$	EC/OC	Biomass Burning Coal Combustion Vehicle Exhausts (EC) Liquid Fossil Fuels (OC)	Active sampling: Directly collected in PM form	Isotope Ratio Mass Spectrometer (IRMS)	IAEA-CH6, IAEA-CH7, USGS24, NBS-19, RM 8573, RM 8542 ^[d]
$\delta^{87}\text{Sr}$	Sr	Coal Combustion Vehicle Exhausts		1. MC-ICP-MS	Sr: NBS987; Pb: NIST SRM981;
$\delta^{144}\text{Nd}$	Nd	Non-exhaust emissions	Active sampling: Directly collected in	2. Thermal ionization mass spectrometer	Nd: JMC, JNdi-1, La Jolla
$^{207}\text{Pb}/^{206}\text{Pb}$	Pb	Natural Soil Ore-related Emissions (Pb & Sr) Waste Incinerator (Pb&Nd)	PM form	(TIMS) 3. ICP-MS (only Pb)	Standard ^[e]
$\delta^{30}\text{Si}$	Si	Biomass Burning Coal Combustion (Si, Fe, Zn) Natural Soil			Si: NIST-SRM-8546, IRMM- 017; LVLK-132; Cu: ERM-
$\delta^{56}\text{Fe}$	Fe	Vehicle Exhausts	Active sampling: Directly collected in	MC-ICP-MS	AE633, ERM-AE647, SRM
$\delta^{65}\text{Cu}$	Cu	Ore-related Emissions	PM form		NIST 976, CAG-Cu; Zn: IRMM-
$\delta^{66}\text{Zn}$	Zn	Non-exhaust emissions (Cu, Fe) Waste Incinerator (Zn)			3702, AA-ETH, SRM-683, JMC 3-0749L ^[f]

[a] The analytical precision (1σ) 0.005‰ – 0.9‰

[b] The analytical precision (1σ) 0.2‰ – 1.5‰

[c] The analytical precision (1σ) 0.15‰ – 0.4‰

[d] The analytical precision (1σ) 0.1‰ – 0.3‰

[e] The analytical precision (1σ) of $\delta^{87}\text{Sr}$: 0.08‰ – 0.3‰, $^{144}\text{Nd}/^{143}\text{Nd}$: 0.004% – 0.005%, $^{207}\text{Pb}/^{206}\text{Pb}$: 0.0004 – 0.0019 or 0.2%–0.5%

[f] The analytical precision (1σ) of $\delta^{30}\text{Si}$ about 0.18‰, $\delta^{56}\text{Fe}$: 0.03‰ – 0.09‰, $\delta^{65}\text{Cu}$: 0.04‰ – 0.09‰, $\delta^{66}\text{Zn}$: 0.03‰ – 0.08‰.